# Conceptual Frameworks Linking Sexual Health to Physical, Mental, and Interpersonal Well-Being: A Comprehensive Systematic Review and Meta-Analysis

**DOI:** 10.3390/bs15121636

**Published:** 2025-11-27

**Authors:** Agnieszka E. Pollard, Ian Cero, Ronald D. Rogge

**Affiliations:** 1Psychology Department, College of Arts and Sciences, University of Rochester, Rochester, NY 14627, USA; aejpollard@gmail.com; 2Psychiatry Department, University of Rochester Medical Center, University of Rochester, Rochester, NY 14642, USA; ian_cero@urmc.rochester.edu

**Keywords:** sexual health, orgasm, well-being, psychological distress, physical health, relationship satisfaction, systematic review, meta-analysis

## Abstract

The current systematic review modified the Enduring Vulnerability Stress Adaptation model of relationship functioning and the Attachment System Activation model of individual functioning to incorporate various aspects of orgasmic functioning within the broader context of sexual health and sexual satisfaction. This provided conceptual frameworks for integrating the findings on a wide range of correlates of orgasms, sexual satisfaction, and other components of sexual health into comprehensive models of individual and interpersonal functioning to guide future research. A systematic search of the *ProQuest*, *PubMed*, and *Web of Science* databases (through September, 2025) for records linking sexual satisfaction with at least one other component of sexual health or at least one correlate (distress, well-being, physical health, relationship satisfaction, attachment avoidance, or attachment anxiety) yielded 3369 unique records, resulting in a final set of 228 records, representing 281 independent (sub)samples and a final combined sample of 248,021 participants. A total of 1201 effects were extracted, yielding 44 meta-analytic effects (using random effects modeling). Path analyses of meta-analytic correlation matrices revealed that dimensions of sexual health (i.e., sexual satisfaction, orgasms, sexual desire, lack of sexual pain, vaginal lubrication) demonstrated unique links to greater health, interpersonal functioning, and individual functioning (i.e., higher psychological well-being, physical health, and relationship satisfaction; lower psychological distress, attachment anxiety, and attachment avoidance). Meta-regression moderation analyses revealed that the effect linking orgasms to higher sexual satisfaction was especially pronounced for women and for individuals in clinical (sub)samples. In addition, the link between orgasms and lower distress was especially pronounced for older individuals. The findings were limited by the cross-sectional nature of the vast majority of the findings (96%), leaving the directions of causality unclear. Taken together, these results highlight the central role that sexual health might play in individual and relationship health, supporting the proposed conceptual models and highlighting directions for future research.

## 1. Introduction

Sexual behavior has long been discussed as one of the most basic human physical and psychological needs ([248]). Consistent with this, previous work has emphasized the importance that sexual health plays in romantic relationships (e.g., [131]; [327]), as well as in other aspects of life in general (e.g., psychological and physical well-being; e.g., [133]; [230]). This growing body of work has developed a multidimensional conceptualization of sexual health which includes not only sexual satisfaction but also orgasmic functioning, sexual desire, vaginal lubrication, lack of sexual pain, and erectile functioning. Given the high value individuals place on orgasms ([276]; [361]; [372]), one line of research has focused specifically on links between orgasmic functioning and well-being, underscoring the importance of orgasms for sexual satisfaction, relationship satisfaction, physical health, mental health, and life satisfaction (e.g., [4]; [42]; [48]; [131]; [226]; [246]). The current systematic review sought to synthesize and integrate this growing line of research on the salience of orgasms in the lives of individuals, by examining orgasmic functioning within the broader context of multiple aspects of sexual health, thereby highlighting the unique links between each aspect of sexual health and the well-being correlates examined. Although a broad array of studies spanning both the psychology and medical literature have examined the correlates of sexual health (see Table 1 and Table 2), the vast majority of this work was atheoretical in nature. In addition, the correlates of orgasms and other aspects of sexual health have generally taken a secondary or even tertiary role in the focus of the analyses presented (often appearing only as a handful of undiscussed correlations in a much larger correlation matrix). The current review therefore sought not only to synthesize this vast array of previous studies, but to also develop a theoretical framework to help conceptually integrate previous findings and guide future work in this area. The current review extended previous research further by using a meta-analytical framework to synthesize and integrate previous findings, thereby allowing us to examine moderators of the links between various aspects sexual health and various forms of well-being. Given the gender disparities uncovered in the field of sexual health (e.g., [14]; [32]), gender was tested as a moderator. As sexual desire and performance vary with age (e.g., [148]; [356]), age was also tested as a moderator of the salience of sexual health across the lifespan. Finally, given the reduced levels of sexual health observed among individuals with mental or physical disorders (e.g., [18]; [75]), the type of population sampled within each study (i.e., clinical or nonclinical) was also examined as a moderator.

### 1.1. Conceptualizing Sexual Health

**Sexual Health as a Central Process.** Defining sexual health is a complex task, as it encompasses physical, emotional, mental, and social well-being ([388]). As sexual health is not merely the absence of dysfunction, but a holistic experience of well-being, it remains imperative to investigate sexual functioning in its scope beyond dysfunctionality. As such, pleasurable sexual experiences do not focus only on achieving orgasms or other physiological factors such as vaginal lubrication, erectile function, or lack of pain, but also include emotional components such as sexual desire, sexual arousal, and sexual satisfaction. Consistent with this, internationally validated sexual health measurement scales such as the Derogatis Sexual Functioning Inventory (DSFI; [112]; [111]), Female Sexual Function Index (FSFI; [310]), and International Index of Erectile Function (IIEF; [311]) provide the tools for researchers to embrace a more diverse and multivariate conceptualization of sexual health (i.e., including sexual satisfaction, orgasmic functioning, sexual desire, vaginal lubrication, lack of pain, and erectile functioning). Thus, although a large body of work has incidentally examined sexual dysfunctions as secondary symptoms of physical health issues (such as cancer, obesity, or psychiatric diagnoses; e.g., [70]; [88]; [182]), the current review applies a novel lens to this literature by examining sexual health as a critical aspect of individual and interpersonal well-being that spans a wide range of populations (both clinical and non-clinical) as well as a wide range of contexts (representing a set of dynamic processes rather than just secondary symptoms).

**Orgasms as One Component of Sexual Health.** A growing body of work has more specifically focused on orgasmic functioning as a key aspect of sexual health (e.g., [4]; [42]; [48]; [131]; [226]; [246]). In fact, there is evidence suggesting that orgasms are still widely perceived as a key goal, if not the ultimate goal of sexual activity (e.g., [276]; [361]; [372]), supporting their use as a marker of sexual health. As a counterpoint to those findings, an emerging area of research has also noted limitations of placing too much weight on orgasms alone, pointing out that: (1) not all orgasms are pleasurable (e.g., [73]), (2) sexual pleasure consists of other crucial elements as well (e.g., [73]), (3) satiating sexual pleasure can be achieved without orgasming (e.g., [276]), and (4) some individuals even experience orgasms despite engaging in unpleasurable or coerced sexual activity (e.g., [73]). Thus, to integrate these perspectives, when reviewing the literature on how orgasms are linked to physical, mental, and relationship functioning, it is critical to examine orgasmic functioning within the broader multivariate context of sexual health. Thus, the current review examined orgasmic functioning alongside the other key aspects of sexual health, including sexual satisfaction, desire, lack of pain, vaginal lubrication, and erectile function. This allowed us to uncover the unique links between each aspect of sexual health and various aspects of individual functioning.

**Modeling Sexual Health.** Although scales like the FSFI and IIEF served to operationalize as many as eight distinct facets of sexual health across men and women, those aspects of sexual health are fundamentally interrelated and therefore notably correlated with one another. Thus, in examining the links between components of sexual health and various aspects of well-being, the current review sought to incorporate those interrelations within multivariate path models. Specifically, the current review conceptualized sexual satisfaction as an overarching construct and the other components of sexual health as key contributors to that global evaluation.

### 1.2. Overview of Research on Orgasms

To ground the current review within a broader perspective of the diversity of studies that have examined orgasmic functioning, the following sections provide a brief overview of the broader orgasmic functioning literature. The following overview also allows us to briefly review many of the studies that helped to shape the various fields of research on orgasms but did not meet the criteria to be included in the current meta-analysis. Finally, it allows us to describe some specific studies to provide a deeper sense of the methods commonly used across these studies. Within the literature, experiencing orgasms has been assessed and quantified in three main ways: (1) frequency of orgasms within a recent time frame (e.g., in the last month), (2) consistency of achieving orgasms from sexual activity (e.g., proportion of sexual encounters or activity that results in orgasms), or (3) a group contrast between individuals experiencing orgasms and those not experiencing orgasms (either at a lifetime level or from recent sexual activity). Given these differing operationalizations, we will use the terms “orgasmic functioning,” “orgasms,” and “experiencing orgasms” as umbrella terms to represent all three conceptualizations.

**Orgasm Gender Gap.** When examining the salience of orgasms in the lives of men and women, the first issue that needs to be acknowledged is that the ability to achieve orgasms is quite different across the sexes. For example, a nationally representative sample of Australians suggested that 95% of men achieved orgasm in their most recent sexual encounter, compared to only 69% of women ([307]). Consistent with this, data from a national probability sample of 3159 individuals in the United States (the National Health and Social Life Survey) suggested that although 75% of men reported “always” having an orgasm during sexual activity with a partner, only 29% of women reported the same level of consistency ([224]). These findings were echoed in a sample of 833 college students with 28% of women and only 3% of men reporting never orgasming with partners, and another 23% of women and only 3% of men reporting only “sometimes” orgasming ([370]). In fact, estimates from a national sample of over 52,000 adults suggest that 95% of heterosexual men in the United States are able to experience orgasms from sexual activity compared to only 65% of heterosexual women ([147]). Women experience orgasms significantly less frequently than men in casual sexual encounters as well as in committed relationships (e.g., [14]; [32]; [147]). Although they have triggered debate within the field (e.g., [49]; [230], [231], [232]; [298], [299]), a number of studies have even suggested that gender differences might appear most pronounced (and show particularly strong associations) for specific types of orgasms from specific forms of sexual activity (e.g., penile-vaginal intercourse—PVI—without clitoral stimulation; e.g., [32]; [47], [48]; [50]; [93]; see [43] for reviews). Despite results suggesting that men and women report comparable levels of overall satisfaction with their sex lives (e.g., [327]), similar gender differences to those observed with orgasms have emerged for reports of sexual desire (e.g., [306]) and sexual pain (e.g., [310]; [335]). Recent results in nationally representative samples continue to support an orgasm gender gap, potentially due to lower frequencies of clitoral stimulation in heteronormative sex ([11]; for a review see [12]).

**Orgasms as Health Correlates.** In the medical and treatment literature, the frequency and the ability to achieve orgasms have often been conceptualized as a health correlate. For example, in depression and depression treatment research, difficulty or inability to achieve orgasms has often been studied as a correlate of depressive symptoms (e.g., [75]; [219]; [225]), as well as a side effect of antidepressant medication (especially selective serotonin reuptake inhibitors; e.g., [31]; [209]; [216]). Orgasmic functioning has also been examined as a secondary symptom in studies of Parkinson’s disease (e.g., [71]; [235]), multiple sclerosis (e.g., [144]; [316]), cancer (e.g., [54]; [142]), chronic pain (e.g., [21]; [81]), and other medical disorders that may impact sexual functioning (e.g., [18]; [37]; [69]; [343]). Although not the primary focus of these studies, this has yielded a body of work linking the symptom of orgasmic difficulties to depressive symptoms.

**Difficulty with Orgasms as a Disorder.** In addition to exploring orgasm difficulties as a secondary dysfunction associated with existing medical disorders, they have also been investigated as primary sexual dysfunctions (e.g., [259]). The 5th edition of the Diagnostic and Statistical Manual of Mental Disorders ([8]) lists a number of sexual dysfunctions, including those of arousal, drive, and pain. It specifically includes three disorders of orgasm functions: Female Orgasmic Disorder, Delayed Ejaculation, and Premature (Early) Ejaculation ([8]). While there is some literature exploring the nature, presentation, and etiology of specific sexual dysfunctions (e.g., [46]; [139]; [164]; [183]; [220]), most research in this area aggregates sexual dysfunctions when examining their correlates (e.g., [121]; [150]) rather than examining specific (e.g., orgasmic) sexual dysfunctions separately. That literature has robustly demonstrated the negative correlates of such disorders for individual and interpersonal functioning, highlighting importance of healthy sexual functioning in daily life (e.g., [9]; [140]; [190]).

**Orgasms as a Relationship Process.** A small but growing body of literature has begun to examine sexual behavior as a key process in romantic relationships. Findings across this body of work suggest that frequency of orgasms is positively linked to other aspects of sexual health (e.g., greater sexual desire and satisfaction, less time needed to achieve sexual arousal; e.g., [44]; [186]). Studies have also linked orgasms to greater relationship quality and marital satisfaction (e.g., [52]; [93]; [168]; [186]; [314]) and to greater relationship investment (e.g., [131]). Although a majority of these studies have focused specifically on the importance of women’s orgasms, a handful of studies have linked higher rates of orgasm consistency and frequency in men to higher levels of sexual satisfaction (e.g., [48]; [265]), relationship satisfaction (e.g., [48]; [147]), and even lower mortality (e.g., [4]; [289]).

**Orgasms as a Source of Well-Being.** A promising vein of research focused primarily on orgasms in the lives of women has begun to examine links between sexual activity and individual well-being. For example, women who reported greater frequency of orgasms reported higher self-esteem (e.g., [186]), lower subjective stress (e.g., [33]; [281]), more effective life coping skills (and lower use of maladaptive coping strategies; e.g., [35]; [50]). Inability to achieve orgasms on the other hand has been linked to higher attachment anxiety and lower overall satisfaction with life, health, and romantic relationships (e.g., [94]). The vast majority of the studies examining these links have been cross-sectional in nature, leaving potential directions of causality unclear. Results from one of the few studies to have employed a longitudinal design suggested that women’s enjoyment of sexual intercourse and men’s frequency of sexual intercourse was linked to greater longevity (i.e., longer life spans; [282]).

### 1.3. Sexual Health Correlates

Extending the review of research focused on orgasmic functioning, the remaining aspects of sexual health have demonstrated similar links to physical and mental health as well as interpersonal functioning. For example, due in large part to the widespread adoption of the FSFI and IIEF within the medical literature, sexual satisfaction, sexual desire, a lack of sexual pain, and vaginal lubrication have each demonstrated links to lower psychological distress (e.g., [234]; [275]), lower attachment anxiety and attachment avoidance (e.g., [295]; [363]), greater well-being (e.g., [15]; [106]), greater physical health (e.g., [210]; [264]), and greater relationship satisfaction (e.g., [338]; [384]). This large body of predominantly cross-sectional findings therefore highlights the importance of examining the correlates of orgasms within the broader context of multiple forms of sexual health.

### 1.4. Organizing Conceptual Framework

Much of the work examining orgasms has adopted a more clinical or practical approach, examining the correlates of orgasms to inform the treatment of medical and/or sexual disorders. The bulk of the work in this area has therefore been atheoretical by design. To begin to integrate this growing body of work into a coherent theoretical framework of both individual and relationship functioning, the current review drew upon the Enduring-Vulnerability Stress-Adaptation model of relationship functioning (EVSA; [204]) as well as the Attachment System Activation model (ASA; [326]). These models therefore informed the selection of correlates to be examined within the meta-analysis.

**Enduring-Vulnerability Stress-Adaptation Model**. In their EVSA model, [204] ([204]) highlight three key sets of processes that interact to shape the course of relationships over time: (1) enduring vulnerabilities (i.e., jagged edges of their personalities that individuals bring with them into relationships such as personality traits and attachment orientations), (2) stressful events (i.e., external events that could impact a relationship such as getting fired, illness, and conflict with family), and (3) adaptive processes (i.e., the adaptive and maladaptive dyadic processes that couples engage in response to life stressors, including constructs such as emotional support, negative conflict, and forgiveness; see Figure 1A for the proposed conceptual model modifying the EVSA to include sexual health). Within the EVSA model, enduring vulnerabilities can directly affect how partners interact with one another within the relationship (influencing dyadic adaptive processes), how the partners adapt in stressful situations (potentially interacting with life stress to exacerbate its impact on relationships), and can even serve to generate stressful life events for the couples to navigate. Stressful events are conceptualized as potentiating events that exert pressure on couples, forcing them to engage their emotion regulation and dyadic coping skills in response. Finally, the dyadic adaptive processes are viewed as most proximally linked to relationship quality, with healthy patterns of interaction helping to buffer relationships from the adverse effects of stress and enduring vulnerabilities, whereas maladaptive patterns are posited to exacerbate those negative effects. Notably, within the EVSA model, enduring vulnerabilities and maladaptive dyadic processes are presumed to require triggering by stressful life events to exert their full influence on relationship quality.

Extending this model to focus specifically on sexual functioning (see Figure 1A), enduring vulnerabilities would likely include a diverse array of constructs such as comfort and knowledge of own body, biological difficulty in achieving orgasms, attitudes toward sex, and attachment orientation, as those trait-like qualities could not only influence dyadic behavior (e.g., affecting how individuals approach sexual activity and sexual communication), but could also serve to generate stress within the relationship (e.g., generating tension over differing views and expectations surrounding the sexual component of their relationships, generating disappointing sexual/intimate encounters). In the context of the EVSA model, failure to have an orgasm within a specific sexual encounter could be conceptualized as a stressful event, potentially interacting with enduring vulnerabilities like sexual expectations, and triggering the need for adaptive processes like greater sexual responsiveness, sexual communication, as well as compassion and empathy. The EVSA model would therefore suggest that adaptive processes like strong emotional support and healthy sexual communication could at least partially buffer relationships from the adverse effects of difficulties with orgasms, whereas maladaptive processes (e.g., withdrawal, avoidance, hostile behavior) would likely exacerbate those adverse effects. In fact, within the context of the EVSA model, the ability to consistently achieve orgasms during sexual activity with a partner could also be modeled as another adaptive process, thereby serving to buffer the relationship from stressful events or the jagged edges of both partners’ personalities. As the need for feeling connected to others has been conceptualized as a fundamental human need ([100]), we conceptualized romantic relationship quality as a key marker of this basic need. As a result, we posited that relationship quality would serve as the most proximal factor (and therefore likely the main mechanism) linking sexual relationship processes to individual well-being.

**Attachment System Activation Model**. To link the current investigation to another robust model in the field of couple’s research, the current study also conceptualized the ASA ([326]) as a more fine-grained model informed by the EVSA model. The ASA model (Figure 1B) expands upon [38]’s ([38], [39], [40], [41]) attachment theory by applying it to adult romantic relationships. Specifically, the ASA model highlights three main components: (1) appraisal of a threat (i.e., something causing stress to a relationship or triggering attachment insecurities), (2) individual attachment insecurities (most commonly conceptualized as attachment avoidance and attachment anxiety), and (3) reactive strategies (i.e., behavioral responses to a threat). Thus, within the ASA model, attachment insecurities are not conceptualized as directly impacting relationship quality. Instead, some sort of threat is required to activate the attachment system, triggering specific behavioral reactions involving either deactivating (e.g., withdrawal, emotion suppression) or hyperactivating (e.g., hypervigilance, rumination) strategies. Given this conceptualization, we see links between the ASA model and the EVSA model, as the ASA model takes a specific enduring vulnerability of attachment, concentrates on a specific subset of (mal)adaptive processes (i.e., deactivation and hyperactivation strategies), and conceptualizes threats as potentiating the activation of that system in a manner similar to the role of life stress in the EVSA model. Although sexual functioning could be considered an independent system from the attachment system, we assert that the ASA model could be meaningfully applied to sexual functioning. From a sexual functioning perspective, orgasm difficulties and a failure to achieve an orgasm during a sexual encounter with a partner could be conceptualized as a possible threat (i.e., a failure of that masculinity achievement, see [74]), activating the attachment system, and thereby promoting deactivation and hyperactivation strategies. Thus, a partner with high levels of attachment avoidance (i.e., feeling uncomfortable with emotional disclosure and intimacy) might withdraw when faced with orgasm difficulties, avoiding communication as well as avoiding further sexual intimacy. Such deactivation strategies would in turn impact relationship quality and eventually individual functioning over time. In contrast, a partner with high levels of attachment anxiety (i.e., a general tendency to feel that partners do not love you as much as you love them leading to excessive preoccupation and worry) might hyper-engage their partner when faced with orgasm difficulties, ruminating over the issue, and possibly becoming demanding and requiring excessive validation. Given this conceptual framework, the current review focused on examining an array of correlates of orgasms spanning the various components of these models (see bolded examples in Figure 1).

### 1.5. Previous Reviews

**Narrative reviews.** The current comprehensive literature search (described below) failed to identify any published meta-analytic systematic reviews with a comparably broad focus (i.e., exploring associations amongst components of sexual health and between those components and a range of factors representing individual and relationship functioning). However, the current literature search did uncover a number of published narrative reviews that focused on related topics. For example, [380] ([380]) published a narrative review of 74 articles examining sex during pregnancy, however none of their articles overlapped with the current sample of 228 records. Similarly, narrative reviews of female sexual functioning in old age ([387]; 3 of its 58 articles overlapped with the current review), the health benefits of various sexual activities ([45]; 9 of its 174 studies overlapped), the psychological and interpersonal correlates of sexual dysfunction ([53]; 1 of its 364 articles overlapped), the links between sexual activity and both physical and mental health ([230]; 5 of its 74 articles overlapped), and links from women’s orgasms and well-being ([114]; 6 of its 85 studies overlapped) demonstrated similar low levels of overlap with the current review. Notably, although [260] ([260]) published a narrative review of 323 articles examining women’s orgasms, the extremely broad scope of that review represented such a distinct focus that none of its articles overlapped with those in the current meta-analysis.

**Systematic reviews.** The current comprehensive literature search also uncovered a number of relevant systematic reviews that have been published in the last 5 years. Given the distinct and slightly more narrow conceptual focuses of these reviews, they only demonstrated nominal overlap with the current review as they examined: (1) etiological factors shaping female sexuality ([278]; 3 of its 21 studies overlapped), (2) predictors of sexual satisfaction ([303]; 6 of its 109 provided citations overlapped), (3) factors linked to sexual functioning in people living with HIV ([184]; none of its 26 studies overlapped), and (4) factors linked to distress over lower sexual functioning ([342]; 1 of its 19 studies overlapped). Finally, the current literature search uncovered a single meta-analytic systematic review with a related focus: examining various aspects of sexual communication and their links to sexual and relationship functioning ([245]). However, given its primary focus on aspects of sexual communication, only 5 of its 93 studies overlapped with the current review.

Taken as a set, this overview of reviews of sexual health (and their markedly low levels of overlap with the current review) suggests that the current review offers a unique contribution to the current literature, integrating findings across a wide range of disparate fields and testing novel path models to evaluate unique (i.e., incremental) links from the various aspects of sexual health to the correlates examined. It also represents the first review to systematically examine the correlates of orgasmic functioning within the broader context of a multivariate conceptualization of sexual health, evaluating the unique links between orgasms and both individual and interpersonal functioning after controlling for other key aspects of sexual health.

### 1.6. Present Meta-Analysis

With the aim of integrating and summarizing research across various domains of functioning, the current study draws from literature in social psychology, clinical psychology, and the medical literature to provide a meta-analytic review of associations between orgasms, and individual and relationship functioning. Based on the previous literature, we anticipated that all six of the sexual health dimensions (i.e., sexual satisfaction, orgasms, sexual desire, lack of pain, lubrication, erectile function) would show significant bivariate meta-analytic correlations to lower distress, greater well-being, better physical health, higher relationship satisfaction/quality, lower attachment anxiety, and lower attachment avoidance. To extend that bivariate literature, our hypotheses focused on a set of multivariate path analyses examining unique links between aspects of sexual health and the correlates. Notably, the multivariate question of unique or incremental predictive validity among aspects of sexual health (when examining links to correlates) represents an entirely novel contribution to the field, as it had not been comprehensively explored across the 1262 full-text records screened for the current meta-analysis, nor within the 228 records within the current meta-analysis. As seen in Figure 2, we conceptualized the more focused indicators of sexual health (orgasms, sexual desire, lack of pain, lubrication, & erectile function) as individual components feeding into sexual satisfaction. Thus, higher levels of functioning on each of those key components were hypothesized to uniquely promote greater sexual satisfaction (i.e., incrementally contributing to greater positive global evaluations; Hypothesis 1). As seen in Figure 2A, for four of the correlates examined (i.e., distress, well-being, physical health, relationship satisfaction), we hypothesized that sexual satisfaction would demonstrate strong proximal links to those more global forms of functioning. Thus, we hypothesized that even after controlling for the other forms of sexual health, higher sexual satisfaction would predict lower distress (Hypothesis 2A), greater well-being (Hypothesis 2B), better physical health (Hypothesis 2C), and higher relationship satisfaction/quality (Hypothesis 2D). Those hypotheses thereby propose that the more specific indicators of sexual health would be indirectly linked to the individual global functioning correlates via their links with sexual satisfaction (Hypothesis 3). After controlling for those indirect links, we further hypothesized that those more specific components of sexual health (including orgasmic functioning) would show unique predictive links to this set of correlates, further highlighting the central nature of sexual health in the lives of individuals (Hypothesis 4). As attachment avoidance and anxiety reflect more stable characteristics that individuals bring into sexual relationships and encounters (consistent with the EVSA and ASA models), we used a different model to examine associations between those correlates and sexual health. As shown in Figure 2B, we hypothesized that greater attachment insecurities would be linked to lower levels of sexual functioning on the more focused indicators (Hypothesis 5), thereby showing indirect links to lower sexual satisfaction (Hypothesis 6). Even after controlling for those indirect paths involving the more focused aspects of sexual health, we finally hypothesized that greater attachment insecurities would be uniquely linked to lower sexual satisfaction (Hypothesis 7).

**Potential Moderators.** Drawing from previous research, the current study examined a set of moderators primarily focused on identifying subpopulations more likely to experience difficulties achieving orgasms, hypothesizing that the associations with experiencing orgasms would be stronger in those challenged populations. Given the marked gender disparities already uncovered in this field of research (e.g., [14]; [32]), gender of participants was used as the primary moderator, anticipating that the correlations might be stronger in women given their lower rates of achieving orgasms. To allow the analyses to be sensitive to the demographic differences across samples, a number of study-level moderating variables were also extracted and examined. As sex drive and sexual performance vary with age (e.g., [148]; [356]), orgasms might take on substantively different levels of salience for psychological and relationship health across the lifespan. Consequently, the average age within each sample was tested as a possible moderator, anticipating that orgasms might show stronger associations in older individuals. In addition, given the lower levels of sexual health found among individuals suffering from mental or physical disorders (e.g., [18]; [75]), the type of population sampled within each study (i.e., clinical or nonclinical) was also examined as a moderator, anticipating that the associations would be stronger within clinical populations. Finally, to directly examine any potential publication bias, the publication status of each study (i.e., published or unpublished research) was examined as a possible moderator. If publication bias was present, we hypothesized that the association biases would therefore be stronger within the published literature.

## 2. Method

This opening section of the methods provides details of the systematic comprehensive review of the literature following the order established in the PRISMA 2020 checklist.

### 2.1. Eligibility Criteria

Records were eligible for inclusion in analyses based on the following criteria:written in any language that could be translated using AI tools;consisted of human participants only;contained independent samples (i.e., providing effects within a group of participants that have not been previously published in other articles out of that sample);included a measure of sexual satisfaction;Included at least one other dimension of sexual health OR included a measure of individual functioning (i.e., attachment, depression, distress, life satisfaction, loneliness, psychological well-being, negative affect, stress, well-being, vitality) OR relationship functioning (i.e., relationship quality, relationship satisfaction, attachment avoidance, attachment anxiety);provided statistical indices of a link between at least one aspect of sexual health and either another facet of sexual health OR one of the corresponding correlates (individual or relationship functioning). If relevant variables were measured but an effect of their association was not reported, authors of the record were contacted via repeated emails in an attempt to collect the relevant statistic;reported an effect size specifically either in the form of a Pearson’s *r* correlation coefficient, a standardized regression coefficient, or other statistical value from which a Pearson’s *r* correlation coefficient or standardized regression coefficient could be computed (e.g., a 2 by 2 chi-squared, a Cohen’s *d*; see Section 2.7 for transformation formulas used).

### 2.2. Information Sources

A systematic literature search was conducted in accordance with PRISMA guidelines ([262]), using *ProQuest*, *PubMed*, and *Web of Science* for records available through the end of September 2025. Introduction sections and reverse citations of key research articles and review articles were also searched to ensure the completeness of the comprehensive search.

### 2.3. Search Strategy

Given the central nature of sexual satisfaction in our conceptual models to be tested, we searched for articles including the term “*sexual satisfaction*.” In addition, the articles also had to either include components of sexual heath (e.g., *orgasm, orgasm/ic frequency*, *orgasm/ic consistency*, or *orgasm/ic ability*, *orgasmic functioning*) OR keywords representing individual and/or relationship functioning (*depression*, *distress*, *life satisfaction*, *loneliness*, *positive affect*, *negative affect*, *stress*, *well-being*, *vitality, maintenance behavior*, *relationship conflict*, *relationship longevity*, *relationship quality*, *relationship satisfaction*, *relationship stability*, *support*). To ensure the searches would pull records focused on these constructs, we restricted the searches to the titles, abstracts, and keywords of the articles. Records were evaluated if they contained at least one of the sexual health terms and at least one of the terms from the other search categories.

### 2.4. Data Collection Process

All extracted effects were evaluated for directionality to ensure that they were coded in appropriate directions. In the two records that offered similar correlational effects for two separate orgasm dimensions (orgasm consistency and frequency; [185]; [212]), the two effects were averaged to prevent overrepresentation of those samples in the resulting meta-analyzed effects. The data extraction process was conducted and checked independently by two of the authors. Discrepancies were rare (.5%) and were resolved through discussion. A total of 1203 relevant effects were extracted from the 281 (sub)samples.

### 2.5. Data Items

A coding procedure was developed to extract relevant information from each study, including record level- and sample level-characteristics. Record level-characteristics included: record authors, year of publication, record title, journal, and record type (i.e., published and peer-reviewed article, dissertation, thesis, or book chapter). Sample level-characteristics included: sample size for each specific effect extracted, mean age of participants, percentage of male participants, percentage of Caucasian participants, percentage of married participants, country of sample population, and whether the sample was clinical or community based. Effects representing bi-variate associations among the six dimensions of sexual health were extracted whenever possible, as were bi-variate associations between the dimensions of sexual health and the six correlates (psychological distress, psychological well-being, physical health, relationship satisfaction, attachment avoidance, and attachment anxiety). Given the possibility that gender might moderate the associations between aspects of sexual health and well-being, we extracted separate effects for men and women whenever possible, thereby treating male and female respondents as distinct subsamples within the records presenting results by gender. Similarly, as the links between sexual health and individual functioning might differ within clinical and nonclinical populations, we extracted separate effects for those two populations whenever possible. We therefore use the term (sub)samples to refer to the resulting 281 distinct samples identified within the 228 records as some of those represent the full sample of a record and others represent subsamples. For a comprehensive overview of (sub)sample characteristics, see Table 1 and for a full listing of those records see Table 2.

### 2.6. Classification of Variable Domains

**Classification of Sexual Health.** Sexual health dimensions consisted of sexual satisfaction, orgasms, sexual desire, lack of pain, lubrication, and erectile function, and were most commonly measured using (1) the Female Sexual Function Index (FSFI-S; [310]; used in 40% of the records), (2) single items developed for each study (used in 23% of the records), or (3) the International Index of Erectile Function (IIEF; [311]; used in 8% of the records) with the remaining 29% of studies using a method of assessment unique to each study.

Sexual satisfaction was defined as a general satisfaction with sexual activity and overall sexual life (e.g., “*How satisfied have you been with your sexual relationship with your partner?*”). Orgasms were operationalized as a general ability to experience orgasms (i.e., group contrast between people who have had at least one orgasm from people who have not; e.g., “*Within the past 12 months, have you been unable to achieve orgasms?*”), consistency of orgasms (i.e., percentage or proportion of sexual encounters resulting in orgasms; e.g., “*When you had sexual stimulation or intercourse, how often did you reach orgasm (climax)?*”), or orgasm frequency (i.e., count of orgasms experienced during sexual encounters over a specific time frame; e.g., “*On how many days in the last month did you orgasm during sexual activity?*”). Sexual desire was conceptualized as a desire or interest in sexual activity (e.g., “*How would you rate your level / degree of sexual desire or interest?*”). Lack of pain was defined as lack of discomfort or pain during or following vaginal penetration (e.g., “*How would you rate your level / degree of discomfort or pain during or following vaginal penetration*”). Lubrication was defined as the ease and ability to become lubricated or maintain lubrication during sexual activity or intercourse (e.g., “*How difficult was it to become lubricated (“wet”) during sexual activity or intercourse?*”). Erectile function was conceptualized as the ability to become erect or maintain an erection during sexual activity or intercourse (e.g., “*How often were you able to get an erection during sexual activity?*”).

**Classification of Correlates.** As represented in Figure 1, the current literature review aimed to extract correlates representing key components of the EVSA and ASA models. Although we attempted to assess a broader range of correlates (including the relationship factors of negative conflict, social/emotional support, and relationship stability), only seven distinct correlate domains emerged as having been examined within the previous literature: (1) psychological distress, (2) psychological well-being, (3) physical health, (4) attachment anxiety, (5) attachment avoidance, (6) relationship satisfaction, and (7) sexual satisfaction. Although a majority of the (sub)samples used well-validated measures (see the most commonly used measures listed below), 26% of the (sub)samples used single items to assess these correlates.

***Relationship Satisfaction.*** Relationship satisfaction was defined as a general satisfaction or happiness within a romantic relationship, reflecting its overall quality. This domain included a variety of variables fitting this definition, including relationship satisfaction, marital satisfaction, relationship quality, marital adjustment, and dyadic adjustment. Despite the range of construct names, these scales contained extremely similar item content (e.g., “*How satisfied are you with your relationship?*” “*How rewarding was your relationship?*” “*How warm and comfortable was your relationship?*”). The most common measure used to assess relationship satisfaction was the Relationship Assessment Scale (RAS; [177]).

***Psychological Distress.*** Psychological distress was defined as a difficult or negative psychological experience. This domain therefore included the more specific constructs of: depressive symptoms (e.g., the Beck Depression Inventory; BDI; [29]), psychological distress (e.g., the Mood and Anxiety Symptom Questionnaire; MASQ; [376]), anxiety (e.g., Hospital Anxiety and Depression Scale; HADS; [397]), negative affect (e.g., the Positive and Negative Affect Schedule; PANAS; [377]), and stress (e.g., the Perceived Stress Scale; PSS; [86]).

***Psychological Well-Being.*** Psychological well-being was defined as an adaptive or positive psychological experience and therefore included the constructs of: vitality (e.g., the Short-Form Health Survey; SF-36; [375]), positive affect (e.g., the PANAS; [377]), well-being or mental adjustment (e.g., [247]), quality of life (e.g., the Quality of Life scale; QOL; [176]), and life satisfaction (e.g., the Satisfaction With Life Scale; SWLS; [115]).

***Physical Health.*** Physical health was defined as the perceived overall quality of physical health (e.g., “*My health is excellent*”). Thus, records reporting physical health as an orgasm correlate used the Short-Form Health Survey (the SF-36; [375]; or the SF-12; [374]) or the physical health subscale of the World Health Organization Quality of Life Assessment (WHOQoL; [381]).

***Attachment Anxiety and Avoidance.*** All records reporting attachment anxiety and avoidance as orgasm correlates used the Experiences in Close Relationships Questionnaire—Revised (ECR; [145]).

### 2.7. Statistical Analyses

**Effect Measures—Transforming Effects.** A majority of the records presented effects as correlations (75%). When both the orgasm experience and correlates were converted into group contrasts (creating an effect in the form of a chi-squared, 7% of effects), that effect was first converted into a 2 × 2 chi-squared with one degree of freedom (collapsing groups if necessary). That allowed the use of the following formula to transform those values into Pearson’s *r* correlations (see [312] for the *k* correction to the typical formula): *r* = sqrt(χ^2^/*nk*), in which χ^2^ represents the chi-squared value, *n* represents the total number of participants used in the analysis, and *k* represents the ratio of proportions between groups (i.e., individuals able to achieve orgasms vs. individuals unable to achieve orgasms). When an odds ratio value was provided (4% of effects), the following formula was used to transform this value to a standardized regression coefficient: *β* = *ln*(*OR*). All effects presented as regression coefficients (11%) were transformed into correlations: *r_xy_* = *β_x_* × (*SD_x_/SD_y_*). When group means, standard deviations, and numbers of participants were provided (3% of effects), we computed a Cohen’s *d* and then transformed that into a Pearson’s *r* correlation coefficient: *r* = *d*/(*d*^2^ + *a*) where *a* = (*n*_1_ + *n*_2_)^2^/(*n*_1_*n*_2_).

**Synthesis Methods: Meta-Analytic Analyses.** Analyses were conducted using Rstudio v.1.1.453 ([305]) using the *foreign* and *metafor* packages ([304]; [366]). Given the wide variety of sample populations, sample sizes, measurement instruments, and study designs, we used random-effects models to estimate our meta-analytic effects ([34]; [92]). *I*^2^ estimates were used to estimate levels of heterogeneity among effect sizes. Cochran’s *Q* estimates were used to quantify each sample’s weighted contribution to the meta-analysis ([83]). All extracted effects in their forms as correlations were transformed to Fisher’s *Z* values and weighted by sample size before analysis. These effects were then meta-analyzed, and the results were subsequently transformed back into correlations for ease of interpretation ([237]). Meta-analytic effects were interpreted using [85]’s ([85]) correlational effect size guidelines, with *r* = .10 indicating a small effect, *r* = .30 indicating a moderate effect, and *r* = .50 indicating a large effect. The presence of potential outliers was assessed using the *influence.measures* function, which calculates outlier diagnostics (e.g., studentized residuals, Cook’s distances, covariance ratios), and identifies individual effects that are disproportionally influential to the overall effect. These analyses identified a handful of outlying effects. However, as results from analyses with and without the outliers remined relatively unchanged and excluding outliers could introduce additional biases, all reported results were from analyses conducted including that handful of outlying effects.

**Study Risk of Publication Bias Assessment.** *Funnel plot asymmetry* tests were conducted to evaluate the possibility of publication bias in this set of records. Funnel plots were visually inspected for asymmetrical distribution of effects around the funnel plot. This distribution was further tested using *Egger’s regression tests* ([128]). If asymmetry was present, *trim and fill* analyses were conducted to estimate the effect sizes that might have emerged without that bias ([125]). Publication bias was further examined using selection method analyses conducted in Rstudio with functions developed and validated by [252] ([252]). These analyses estimate the relative probability of a contradictory finding (i.e., non-significant or in the opposite direction) being included in the analysis in comparison to the probability of a consistent and significant finding being included. Thus, relative probabilities close to a value of 1.0 would suggest the presence of very little publication bias within the current sample, whereas relative probabilities much lower than 1.0 would suggest publication bias.

**Incremental Prediction Analyses.** To examine the unique links between the various aspects of sexual health and each correlate, path analyses were run on meta-analytic correlation matrices within Mplus 7.11. Although we had planned on including erectile function as a dimension of sexual health in these analyses, there were insufficient studies providing correlations with erectile functioning to create the necessary meta-analytic correlation matrices. This restricted our path analyses to sexual satisfaction and the four remaining sexual health components (orgasms, desire, lack of pain, and lubrication). Sexual satisfaction was typically assessed as a global positive evaluation of individuals’ sex lives and therefore represents an overarching construct to which the other aspects of sexual health contribute. To recognize this within our path models, we allowed the other dimensions of sexual health to predict levels of sexual satisfaction (Figure 2). The correlates representing global functioning (distress, well-being, physical health, and relationship satisfaction) were then modeled as outcomes (Figure 2A), treating global sexual satisfaction as a mechanism linking the more specific components of sexual health to each correlate). The four more specific aspects of sexual health were also allowed to directly predict levels of the correlate, after controlling for: (1) their links to sexual satisfaction, (2) the link between sexual satisfaction and the correlate being examined, (3) the associations among those four more specific indices of sexual health, and (4) the unique predictive links of each of those four indices to the outcome. This allowed our models to estimate the unique predictive associations of each of those indices of sexual health. In contrast, the correlates of attachment anxiety and attachment avoidance were treated as predictors of both the four more focused aspects of sexual health as well as sexual satisfaction (which served as the outcome; Figure 2B). As the path models tested were fully saturated, they yielded a perfect fit.

**Exploring Heterogeneity—Moderator Analyses.** Meta-regression (using mixed effects models to accommodate the heterogeneity of the records) was used to assess the degree to which participant gender, participant age, (sub)sample population (i.e., clinical vs. nonclinical), and publication type moderated the associations between aspects of sexual health and the individual and relationship functioning correlates examined. More specifically, we focused our moderation on associations between aspects of sexual health and the correlates with sufficient numbers of (sub)samples to support the analyses: (1) sexual satisfaction (to examine moderation of the contribution of more specific sexual health factors to overall evaluations of sexual well-being; *k* = 118), (2) psychological distress (*k* = 84), (3) psychological well-being (*k* = 20), and (4) relationship satisfaction (*k* = 51). The moderators were entered into the meta-regressions simultaneously, thereby serving as controls for one another so that the analysis evaluated their unique moderation of the meta-analytic effects.

## 3. Results

### 3.1. Yield of Comprehensive Literature Search

The initial database searches (of *ProQuest*, *PubMed*, and *Web of Science*) yielded a total of 3168 unique records that were screened for eligibility by a minimum of two of the authors based on their titles and abstracts (see Figure 3 for a PRISMA diagram). In order to maximize inclusion of unpublished records, *Google Scholar* was used to conduct a comprehensive reverse citation on some of the most relevant and highly cited articles (included in the current meta-analysis) and on 3 relevant review articles ([45]; [230]; [260]). This yielded another 750 records for a total of 3369 unique records. Screening of the titles and abstracts of those records yielded 1262 full-text articles that were screened by the first and second authors, yielding a final set of 228 unique records representing 281 independently analyzed (sub)samples.

### 3.2. Overview of Records

Table 1 presents an overarching summary of the 281 (sub)samples yielding the effects for this meta-analysis. Table 2 then presents details on each of the 228 records yielding those (sub)samples to ground the systematic review.

**Participant Characteristics.** Given the orgasm gender gap (e.g., [147]; [224]), 65% (134) of the resulting (sub)samples were focused exclusively on examining sexual functioning within women (Table 1). This general trend was balanced by some large-scale records collecting data from both genders or exclusively from men, yielding data from 63,171 men (25.4% of the comprehensive sample of 248,021 unique respondents) for the current meta-analysis. The 281 (sub)samples included in the current meta-analysis were notably international in their scope as the (sub)samples represented over 45 different countries (e.g., China, Italy, Poland, Portugal, Turkey) including one cross-cultural dissertation presenting data from 43 distinct countries ([138]). Thus, likely due to the international adoption of scales like the FSFI and the IIEF, the meta-analytic sample is reasonably globally representative, allowing the results to potentially generalize beyond just the United States and Western Europe. The sample was also reasonably diverse with 64% of respondents (within the 77% of the (sub)samples reporting ethnicity) identifying as Caucasian. Sample average ages ranged from 18 to 74 years old with a weighted average age of 37.1 (SD = 11.0), suggesting that a majority of the respondents were in their 20s, 30s, 40s and 50s. Although 30 of the records (13%) collected data from college students, the vast majority of the samples were drawn from community adults or clinical populations. Consistent with this, 77% of participants were in romantic relationships and the sample average relationship lengths ranged from 1.1 to 34.1 years with a weighted mean of 8.0 years (SD = 5.7; within the 78 records reporting). Taken together, these results highlight a diverse international sample made up largely of young and middle-aged adults typically in long-term romantic relationships.

**Record/Manuscript Characteristics.** Although the 281 subsamples were from articles published in peer-reviewed journals (92.5%), the comprehensive literature search also uncovered relevant unpublished doctoral dissertations (*k* = 12) and unpublished master’s theses (*k* = 9), which were included in the current meta-analyses to help defray the impact of possible publication bias (Table 2). In addition, for a majority of the records identified, the correlations between orgasm constructs and well-being were incidental to the main focus of the papers, with the relevant correlations simply showing up in a study-wide correlation matrix without any associated results narrative. In fact, only 81 records (35%) had the words orgasm or sexual satisfaction in the title. Thus, for A majority of the records in this meta-analytic sample, the significance of the relevant correlations would likely have had little impact on the publication of those manuscripts.

The (sub)samples had been published across a 54-year span, with a majority of the (sub)samples (77%) having been published in the last 15 years (Table 1). The (sub)samples were a fairly even mix of community adults and adults within specific clinical populations. As seen in Table 1, the clinical (sub)samples represented a large variety of different clinical diagnoses (i.e., over 34 distinct diagnoses, including: depression or anxiety, sexual dysfunction, cancer, menopause, and pregnancy). Most of the records included in this meta-analysis were cross-sectional in design (96%; see Table 2), and although a small number of records contained longitudinal designs, only a few reported longitudinal effects between orgasms and relevant correlates such as sexual satisfaction or positive affect (e.g., [55]; [167]). Similarly, only a small fraction of records collected data from both partners within a romantic relationship, and only a small handful of those records (e.g., [156]; [192]; [212]) analyzed the partner data dyadically with approaches like actor-partner interdependence modeling (APIM; e.g., [90]).

**Data Characteristics.** A total of 1201 distinct effects were extracted from the 281 (sub)samples, yielding large numbers of effects (ranging from 169 to 329) for the correlates of relationship satisfaction and psychological distress, and smaller numbers of effects (ranging from 41 to 85) for the correlates of well-being, physical health, attachment anxiety and avoidance (Table 1). Although the comprehensive search screened for the relationship processes of negative conflict behavior and social support as possible orgasm correlates, the searches failed to uncover any records having examined those associations. Given the lower rate of studies examining men’s sexual health, the literature search only uncovered 9 records demonstrating links between erectile functioning and the correlates examined.

### 3.3. Meta-Analytic Correlations

The dimensions of sexual health (i.e., sexual satisfaction, orgasms, sexual desire, lack of pain, lubrication, erectile function) were positively associated with one another (*r* = .265 to .555, *k* = 11 to 138; see bottom of Table 3 for full results). As anticipated, higher levels on each of the dimensions of sexual health were associated with lower levels of psychological distress (*r* = −.276 to −.148, *k* = 11 to 97), higher psychological well-being (*r* = .196 to .343, *k* = 12 to 23), higher physical health (*r* = .221 to .311, *k* = 9 to 12), lower levels of attachment anxiety (*r* = −.242 to −.145, *k* = 2 to 9), lower levels of attachment avoidance (*r* = −.237 to −.043, *k* = 2 to 22), and higher relationship satisfaction. (*r* = .182 to .554, *k* = 11 to 62).

### 3.4. Meta-Analytic Path Analyses

The meta-analytic estimates of the bivariate associations among the constructs being examined (from Table 3) were submitted as correlation matrices to Mplus to evaluate the unique predictive links between each aspect of sexual health and the individual and interpersonal correlates examined. Table 4 and Figure 4 present the standardized path coefficients generated by these models for each of the correlates. Offering support for Hypothesis 1, orgasms, sexual desire, lack of sexual pain, and vaginal lubrication were each uniquely predictive of greater sexual satisfaction (Figure 4A–D) after controlling for their associations with one another. Consistent with Hypothesis 2, sexual satisfaction in turn, uniquely predicted lower psychological distress (Hypothesis 2A, Figure 4A), greater well-being (Hypothesis 2B, Figure 4B), better physical health (Hypothesis 2C, Figure 4C), and higher relationship satisfaction (Hypothesis 2D, Figure 4D), suggesting proximal associations with those indices of global functioning. Asymmetric confidence interval tests suggested significant indirect paths linking more focused components of sexual health to the correlates via higher sexual satisfaction (see Table 4), thereby offering partial support for Hypothesis 3. Thus, greater orgasms, sexual desire, and vaginal lubrication were indirectly linked to better functioning (lower distress and greater well-being, physical health, & relationship satisfaction) through their links to greater sexual satisfaction.

After controlling for those indirect associations, three of the specific aspects of sexual health (orgasms, desire, and lack of pain) demonstrated additional direct links to individual and relationship functioning in the expected directions, offering partial support for Hypothesis 4. Thus, even after controlling for sexual satisfaction and the other aspects of sexual health, greater orgasmic functioning was uniquely linked to three of those four correlates (lower psychological distress, greater physical health, and greater relationship satisfaction) further augmenting its indirect links to those outcomes via higher sexual satisfaction. Similarly, a lack of sexual pain was uniquely linked to slightly lower psychological distress, greater well-being, greater physical health, and slightly higher relationship satisfaction. Finally, sexual desire was uniquely linked to greater well-being, physical health, and relationship satisfaction. After controlling for the other aspects of sexual health as well as indirect links to functioning via sexual satisfaction, multivariate suppressor effects emerged for vaginal lubrication. Thus, higher levels of the residual aspects of vaginal lubrication that were completely independent of levels of orgasms, desire, and sexual satisfaction were linked to slightly lower well-being, physical health, and relationship satisfaction.

Turning to the path models examining attachment insecurities as predictors of sexual health, attachment avoidance was linked to lower levels of all four specific aspects of sexual health (Figure 4E) and attachment anxiety was linked to lower orgasms, greater sexual pain, and lower vaginal lubrication (Figure 4F), offering partial support for Hypothesis 5. As seen in Table 4, asymmetric confidence interval tests revealed significant indirect links between attachment insecurities and lower sexual satisfaction via their links to lower levels of the more specific aspects of sexual health, offering partial support for Hypothesis 6. Even after controlling for those indirect links through specific aspects of sexual health, both attachment avoidance (Figure 4E) and attachment anxiety (Figure 4F) demonstrated additional direct links to lower sexual satisfaction, supporting Hypothesis 7. Taken as a set, these path analysis findings highlight the unique roles that various aspects of sexual health play in the lives of individuals.

### 3.5. Moderation Effects

Moderation analyses were conducted to estimate the moderating effects of gender, (sub)sample population (i.e., clinical vs. nonclinical), age, and publication type on the bivariate associations between specific aspects of sexual health and the three correlates to which sexual health was linked across at least 20 studies (offering sufficient numbers of effects to support these analyses): psychological distress, well-being, and relationship satisfaction. As the specific components of sexual health were conceptualized as contributing to overall sexual satisfaction, moderation analyses were also conducted on those predictive links. Given the broad range of samples, methods, and measures employed across the 281 (sub)samples, the *Q* statistics for the effects examined were all significant, suggesting meaningful amounts of heterogeneity to support moderation analyses. Weighted random-effects *meta-regression* models were run using the *metafor* package in Rstudio to simultaneously test the unique effects of these four moderators on the links between orgasms and each of the outcomes. The terms testing the moderators were all centered on their weighted grand means prior to running the analyses. As shown in Table 5, when tested simultaneously, only a handful of significant moderation effects emerged from these analyses, thereby suggesting that a majority of the meta-analytic effects generalized across these moderators.

**Moderation by Gender.** Despite notable gender differences on orgasmic functioning and sexual desire between the primary genders, gender largely failed to emerge as a significant moderator for all but one of the effects tested. Gender only emerged as a unique moderator of links between orgasms and sexual satisfaction (*β* = −.172, *p* = .001). Thus, although orgasms are linked to higher satisfaction across both primary genders, this effect was predicted to be significantly stronger in women (*β* = .444) than in men (*β* = .444 − .172 = .272), suggesting that orgasmic functioning might be more salient for sexual satisfaction in women.

**Moderation by Age.** After controlling for the other moderators, average sample age emerged as a significant moderator of the links between orgasms and psychological distress (*β* = −.006, *p* = .002). As that predictor was centered at 37.1 years (the weighted mean across all samples), these results predict only a weak association for samples with average ages of 18.1 years (*β* = −.237 + (−19) × (−.006) = −.123) but a notably stronger association for samples with average ages of 57.1 (*β* = −.237 + (20) × (−.006) = −.357). Similarly, age significantly intensified the positive links between: (1) sexual desire and sexual satisfaction (*β* = .006, *p* = .032), (2) lack of pain during sex and sexual satisfaction (*β* = .012, *p* = .019), and (3) lack of pain during sex and relationship satisfaction (*β* = .008, *p* = .025).

**Moderation by Clinical vs. Non-Clinical Population.** Despite spanning over 33 distinct diagnoses, a majority of the conditions represented were more chronic in nature resulting in a shared experience of more chronic levels of impairment. Thus, we treated clinical vs. non-clinical populations as one of our moderators to be tested, collapsing across those individual disorders to focus on how impairment in health and individual functioning might impact the links examined. Population type emerged as a unique moderator of the links between orgasms and sexual satisfaction (*β* = .130, *p* = .001), such that the association was significantly stronger in samples drawn from clinical populations. This suggests that orgasmic functioning might take on particular salience for well-being in clinical populations.

### 3.6. Publication Bias

As seen in Table 5, after controlling for the other moderators, publication status failed to emerge as a significant moderator of the links between specific aspects of sexual health and the constructs with sufficient numbers of effects to support meta-analytic regressions. This suggests that the 9 effects tested (for which publication status could be tested as a moderator) did not significantly differ between published peer-reviewed and unpublished research (sub)samples. Consistent with this, *Egger’s regression tests* only identified significant funnel plot asymmetry for 20 of the 44 effects (see Table 3) and the resulting shifts in meta-analytic effect sizes from *trim and fill* analyses were largely minimal. In fact, the trim and fill analyses yielded unchanged estimates for 4 of those 20 effects and stronger estimates for 15 of them. This is likely a consequence of 7.5% of the (sub)samples being drawn from unpublished sources. It is also likely due in part to the fact that in roughly 65% of the records, the relevant effects being extracted were incidental to the main focus of those manuscripts (typically appearing within a study-wide correlation matrix without ever being discussed). As a result, the significance of those effects would have had no effect on the publishing decisions for those articles. Taken together, these findings converge to suggest that minimal levels of publication bias were present in the meta-analyzed effects.

In contrast, the relative probabilities estimated by selection method analyses (e.g., [252]) provide a note of caution to those broader publication bias findings. The relative probability of a contradictory finding (e.g., non-significant or in the opposite direction) being included in the current review was .75 or greater (suggesting fairly reasonable odds of finding published results that were either non-significant or even inconsistent with the predominant findings) for 27 of the 44 effects. However, the probabilities of contradictory findings being included for the remaining correlates were occasionally lower, suggesting that current meta-analytic effect estimates might have been slightly inflated by the publication (or inclusion) biases.

## 4. Discussion

As research studies from diverse fields have explored the potential benefits of experiencing orgasms and sexual health on physical, emotional, and interpersonal well-being across the last 49 years (often incidentally to the primary foci of those studies), this meta-analysis drew from clinical psychology, social psychology, and medical studies to integrate that vast body of work. Thus, the current literature review resulted in a set of 228 records, yielding 281 (sub)samples and 1201 effects, representing a combined total sample of 248,021 participants. Given the importance that individuals continue to place on orgasms (e.g., [276]), our primary focus was to examine the links between orgasmic functioning and various indices of well-being. However, given the multivariate perspective on sexual functioning that has developed within the literature (e.g., [310]), we took a broad perspective and examined orgasmic functioning as one component within the greater context of sexual health. Consistent with our modified EVSA and ASA models, the meta-analytic findings and subsequent path analyses revealed unique links from the various aspects of sexual health (i.e., sexual satisfaction, orgasms, sexual desire, lack of pain, vaginal lubrication, erectile function) to physical health, individual well-being and relationship well-being. Meta-analytic moderation results further revealed stronger links between orgasms and specific forms of well-being for: (1) women, (2) individuals from clinical populations, and (3) older individuals. As the first published meta-analysis in this area, the review sought to integrate findings from diverse fields of study within the EVSA and ASA conceptual frameworks, providing a clear focus to the review and frameworks to guide future work. The current study further offered a quantitative synthesis of the correlates of orgasms, which enabled us to markedly advance the literature by quantifying the unique associations of various aspects of sexual health with a range of individual and interpersonal correlates.

### 4.1. Implications

**Promising Conceptual Frameworks.** The focus of the current review was conceptually grounded in the EVSA and ASA models (i.e., seeking model-consistent correlates). Although the meta-analytic path models tested fell short of truly testing those more complex models, the current findings demonstrated robust links between sexual health and key constructs from those two models. Thus, the current findings offer a compelling foundation to support using the EVSA and ASA models as conceptual frameworks to guide future work in this area. For example, future studies of romantic relationships would likely benefit from modeling aspects of sexual health as adaptive processes, chronic sexual difficulties as enduring vulnerabilities, and/or failure to achieve orgasm as a stressor within the context of the EVSA model. Similarly, studies focused on the role of adult attachment insecurities could potentially benefit from modeling sexual health difficulties as possible threats that could trigger the activation of the attachment system within the context of the ASA model.

**Sexual Health Benefits Relationships.** Consistent with previous literature, current findings demonstrate that sexual health is linked to overall relationship quality (e.g., [42]; [93]; [131]; [288]; [349]; [384]), highlighting the potential importance of sexual health within romantic relationships for both men and women. Although these findings provide a solid foundation for examining sexual health as a distinct relationship process that could influence relationship quality over time, given the cross-sectional nature of the vast majority of the studies reviewed, future work is needed to explore the direction of those associations. In fact, the EVSA and ASA models highlight a myriad of more specific relationship processes that have yet to be examined and modeled with sexual health (e.g., sexual communication, relationship conflict, partner responsiveness, social support, attributions for partner behavior, mindfulness, psychological flexibility, gratitude, demand-withdrawal, and hypervigilance). Although those relationship processes have yet to be examined in the context of sexual health, they are likely to interact with sexual health to shape the course of relationships. Thus, the conceptual frameworks organizing this review further highlight an array of promising directions for future work.

**Sexual Health Benefits Individuals.** Although sexual behavior is often a dyadic experience, current findings and previous literature suggest that the benefits of sexual health extend far beyond relationship functioning (e.g., [230]). For both men and women, sexual health appears to have important implications for individual functioning, (e.g., [44]; [95]; [133]; [256]; [338]). The current findings suggest that all six dimensions of sexual health are linked to lower psychological distress, higher psychological well-being, and greater physical health. In the case of the correlates of orgasmic functioning, this could be explained in part by findings that sexual activities, and orgasms even more so, release prolactin and oxytocin (e.g., [66]; [217]; [228]; [244]; [259]), hormones which have been shown to demonstrate calming satiation and stress relieving features in both men and women (e.g., [211]; [218]; [230]; [332]; [360]), Thus, it may be interesting and useful to explore the potential stress-buffering benefits of orgasms and other sexual health dimensions in future studies. In the context of the ASA model, healthy sexual functioning, as well as consistent and high-quality experiences of orgasms might also serve as buffers to prevent perceived threats from triggering the attachment system, thereby lowering the stress experienced by individuals with attachment anxieties. Similarly, within the EVSA model, pleasurable sexual experiences might actually serve as an adaptive relationship process that buffers relationships from the jagged edges of individuals’ personalities and from the adverse impact of stressful events, thereby promoting individual well-being by bolstering relationship quality. Future work could explore these various mechanisms linking sexual health and experiences of orgasms to greater individual health.

**Orgasms Matter—Particularly to Women.** Although positive links to orgasms are found for both men and women, it appears that they are especially salient for women. As existing literature highlights, women experience orgasms at notably lower rates than do men (e.g., [14]; [32]; [147]; [369]; [370]). Furthermore, although both men and women experience orgasm sexual dysfunctions, men more typically experience premature or delayed orgasms, in contrast to the complete absence of orgasms that many women experience (e.g., [105]; [188]). Such findings, in combination with the results of the current meta-analysis, suggest that as women experience less frequent orgasms, the links between orgasms and positive correlates might become especially crucial for them. Although not a focus of the current review, a related line of study has demonstrated that greater frequency of women’s orgasms is linked to pleasure-focused sexual education received in childhood or adolescence (e.g., [51]), highlighting possible points of intervention. Thus, future work could examine the more developmental predictors of both women and men developing the skills to have consistent and high-quality experiences of orgasms from pleasurable sexual activity.

**Understanding Vaginal Lubrication**. Consistent with our hypotheses, vaginal lubrication demonstrated: (1) positive bi-variate associations with the other aspects of sexual health, (2) adaptive bi-variate associations with the correlates (e.g., greater well-being, lower distress), (3) unique positive links to sexual satisfaction in the path models, and (4) corresponding indirect associations with lower distress, and with greater well-being, physical health, and relationship satisfaction in those same path models. However, after controlling for those indirect links, suppressor effects (see [240]) emerged in the remaining direct links from vaginal lubrication to three correlates in the path models. Thus, the aspects of vaginal lubrication that were completely unrelated to orgasms, sexual desire, lack of pain during sex, and sexual satisfaction were associated with slightly lower well-being, physical health, and relationship satisfaction. As these suppressor effects are based on residual variance, they should be interpreted with caution as they tend to be less stable and might not continue to emerge with a slightly different set of covariates (see [240] for a discussion of suppressor effects). Having said that, these results suggest that being able to lubricate in the absence of sexual satisfaction or desire might serve as a marker for less traditional sexual attitudes (possibly reflecting a greater comfort and proclivity toward causal sex). Although links between vaginal lubrication and sociosexual orientation are yet to be investigated, having an unrestricted sociosexual orientation (i.e., being more embracing of casual sex) has been linked to lower relationship satisfaction and quality (e.g., [223]; [359]) whereas it has been linked to greater psychological well-being and lower psychological distress after engaging in casual sex ([368]). Given that most effects were drawn from samples of individuals in romantic relationships, it may be possible that having unrestricted sociosexual orientation might have a negative effect on individual and relationship functioning as those individuals found themselves constrained by what were likely to be predominantly monogamous relationships. Thus, future research could explore this phenomenon and examine potential links between vaginal lubrication (and other aspects of sexual health) and sociosexual orientation.

### 4.2. Future Directions

While the current systematic review has unified findings from a diverse set of literature, it has also uncovered a number of areas for future work that have yet to be explored.

**Theoretically Grounded Studies**. Despite a large body of work supporting the current meta-analytic findings, much of that work was more pragmatic in nature (i.e., examining sexual functioning as a secondary outcome in studies of physical illness) than conceptually focused. Thus, the effects extracted from those studies were typically tangential or completely unrelated to the primary focus of the manuscripts and were often not even discussed within the results narratives. As a result, the current findings offer an important first step toward developing a theoretically grounded program of research in this area, highlighting the EVSA and ASA as potential conceptual frameworks for future studies and therefore suggesting a number of directions to be explored in future studies.

**Examining Mediators.** The mechanisms linking sexual health to relationship quality remain unclear. The EVSA model (see Figure 1A) suggests a host of potential adaptive processes that could be examined as possible mechanisms in future studies (e.g., sexual communication, relationship conflict, demand-withdraw behaviors, partner responsiveness, social support, attributions for partner behavior, mindfulness, and psychological flexibility). Given the atheoretical nature of the vast majority of the research on sexual health, the links between sexual health and these relevant relationship processes have yet to be examined, much less treated as possible mechanisms within larger models of relationship functioning. Future studies could therefore conceptually extend this work by examining models and specific constructs or processes informed by the EVSA or ASA models. For example, positive, consistent, and high-quality sexual experiences might strengthen romantic relationship quality by promoting more compassion, responsiveness, and emotional support within those relationships. Of course, such experiences could even be modeled as a mechanism. For example, a future study could examine how an enduring vulnerability like negative body image might adversely impact romantic relationship quality by lowering the quality and consistency of pleasurable sexual experiences. The EVSA model therefore provides a framework for integrating the correlational findings on orgasms into a variety of conceptual models to be tested.

**Biological Mechanisms.** Although somewhat outside of the scope of the current review, a growing body of work has uncovered possible neurochemical mechanisms linking orgasms to emotional bonding (for reviews and greater details see [341]; [322]). Thus, studies have shown that around the time of orgasm, there is a cascade of changes in cerebral blood flow in the brain including deactivation in left prefrontal cortex and left temporal lobe (e.g., [153]; [152]; [180]; [233]), as well as activation in the cerebellum ([153]; [180]; [259]; [233]), right prefrontal cortex ([180]; [351]), and hypothalamus ([213]; [259], [260]). Specifically, activation in the hypothalamus results in the release of oxytocin, referred to as the “feel good hormone” which facilitates social bonding in both men and women (e.g., [66]; [68]; [269], [268]; [274]; [293]; [360]). Since evidence suggests that oxytocin is fundamental in bonding, this could help explain how couples who have orgasms together feel closer. Thus, in addition to process-oriented models examining psychological and interpersonal processes as mechanisms, future work could extend the current findings by also clarifying and quantifying the biological links between sexual health and well-being.

**Examining Moderation**. Another conceptual possibility suggested by both the EVSA and ASA models is that sexual health might function as a moderator within models of relationship and individual functioning. For example, drawing from the EVSA model (Figure 1A), a future study could examine how experiencing consistent and high-quality orgasms or pleasurable sexual activity (as a dynamic state-like process) might serve as an adaptive relationship process, buffering those relationships from the adverse effects of enduring vulnerabilities and stressful life events. Of course, more pervasive sexual health difficulties (e.g., chronic sexual pain, chronic difficulties with lubrication/erection, or being completely anorgasmic at the more stable trait-level) could be conceptualized as an enduring vulnerability that could create stress within the relationship and could shape the tone of other dyadic processes (i.e., moderating the impact of conflict and support behaviors). Drawing from the ASA model (Figure 1B), future studies might examine how experiencing consistent and high-quality orgasms, or pleasurable sexual activity, might buffer individuals from perceived threats triggering or activating their attachment systems, thereby ameliorating the impact of attachment insecurities on relationship and individual functioning. Within that same framework, chronic and pervasive difficulties with sexual health could be expected to potentially make individuals more reactive to perceived threats, lowering the threshold for attachment system activation and prompting greater levels of deactivating (e.g., withdrawal, avoidance, denial) and/or hyperactivating (e.g., demand, rumination, hypervigilance) behaviors, particularly for individuals with greater levels of attachment insecurities. At more of a dynamic state or event level, having negative sexual experiences or sexual health difficulties during a specific intimate encounter could serve as a threat to the relationship within the ASA framework, triggering hyperactivating and/or deactivating behaviors by activating an individual’s attachment insecurities. Extending that logic back to the EVSA framework, future studies could even explore how the quality of sexual health and/or consistency of pleasurable sexual experiences might interact with other relationship processes (e.g., conflict, support, responsiveness) to help shape the course of romantic relationships over time.

**Examining Predictors of Sexual Health**. Finally, the EVSA model sheds light on how more stable and enduring aspects of individuals can be incorporated into comprehensive models of functioning in future studies. Although the current review focused on attachment insecurities as the only enduring vulnerability examined, previous work has examined a variety of predictors of sexual health including enduring traits like sexual education (e.g., [137]; [51]), negative body image (e.g., [120]; [300]), knowledge of own body (e.g., [370]), and sexual attitudes like erotophilia (e.g., [172]; [185]), sociosexual orientation (e.g., [339]; [365]; [386]), and sexual sensation seeking (e.g., [56]). The EVSA conceptual framework therefore offers researchers a method of integrating those predictive links to sexual health into broader models of individual and relationship functioning. Future studies could therefore build on the current findings and the broader predictive findings within the sexual health literature by expanding beyond just running analyses to sexual health. Specifically, future studies could examine how a broad array of enduring vulnerabilities might not only (1) predict the quality of sexual health and consistency of pleasurable sex or orgasms, but also (2) might generate stressful events for couples to navigate, and (3) interact with stressful life events to influence both sexual health and other relationship processes to shape the course of relationships over time.

**Tracking the Impact of Sexual Health Over Time.** While much research has been conducted on sexual health dimensions and its correlates, the vast majority of this work (96% of the records) has been cross-sectional (with none of the records predicting residual change to ensure that baseline associations would not inflate prediction) leaving the directions of causality unclear. Longitudinal research would greatly inform this field of study. Specifically, analyses in multi-wave longitudinal designs could clarify directions of associations. These designs could include short-term intensive studies such as daily diaries or ecological momentary assessments to examine immediate and daily effects of sexual health dimensions. Previous findings within a daily diary study of 96 couples have suggested that sexual activity leaves a lingering “afterglow” of positive effect on sexual and marital satisfaction for roughly 48 h ([254]). In one of the few records to examine orgasms on daily basis, [55] ([55]) collected 36 weeks of daily diary assessments from 58 women. Their lagged analyses supported reciprocal links between orgasming and positive mood even after controlling for rates of intercourse and physical affection, thereby providing initial evidence of bi-directional causality for those two constructs. Future work could examine the daily correlates of sexual health dimensions with the full range of correlates examined in this meta-analysis as most of those links remain largely unexamined. Future work could also extend the timeframe of diary assessments by using weekly diaries, potentially capturing slightly more lasting effects on relationship and individual functioning. In addition to tracking daily or weekly correlates of sexual health dimensions, long-term designs (e.g., spanning months or years) could also prove informative for examining how this aspect of a couple’s sexual relationship might interact with other relationship processes over the broader course of romantic relationships. For example, it would be interesting to track sexual behavior and orgasms along with other common relationship processes (e.g., social support, negative conflict behavior, forgiveness, aggression) in newlywed couples over the first few years of marriage. As a small but growing body of studies have demonstrated that sexual activity and orgasms offer unique predictive variance (e.g., [93]), it is likely that they could play unique roles across the early years of marriage. Thus, a newlywed couple engaging in high levels of sexual activity in which one of the partners experiences a low rate of sexual satisfaction, desire, or orgasms could very well have a very different trajectory of marital functioning than a couple engaging in lower levels of activity but with a far higher satisfaction, desire, or orgasm rate as a result of those activities. This example highlights how orgasms and other aspects of sexual health might serve to moderate or interact with other relationship processes like sexual activity, physical affection, emotional support and even conflict behavior. As a result, such studies would help clarify the impact of possible sexual health gaps on relationship functioning during that high-risk stage of early marriage.

**Examining Orgasm Specificity.** A series of studies (e.g., [45]; [48]) has focused on emphasizing the distinct differences between types of orgasms (i.e., vaginal orgasms without clitoral stimulation compared to orgasms with clitoral stimulation) and their sources (i.e., through penile-vaginal intercourse (PVI), partnered masturbation, or solitary masturbation). For example, some findings have suggested that PVI frequency may be a stronger predictor of greater sexual satisfaction, relationship satisfaction, mental health, and life satisfaction (e.g., [48]). Similarly, orgasms from PVI without clitoral stimulation have shown strong links to greater positive affect (e.g., [347]). However, these studies have been criticized for potential deficits in scientific rigor, theoretical grounding, and replication by independent researchers ([230], [231]; [298], [299]; [349]). The emphasis on distinguishing between type of orgasms has also been criticized as lacking utility given that the clitoral structure envelops the vaginal opening and is therefore not only stimulated via PVI regardless of direct stimulation of the clitoral glans, but also the resulting orgasms are largely indistinguishable for most women (e.g., [230], [232]; [298], [299]). The exclusive focus on PVI within this line of work also excludes the study of sexual health in sexual and gender minority groups. Given these concerns, the vast majority of the research in this area has yet to fully explore this specificity. Consequently, there were too few published results to support the estimation of meta-analytic effects for specific types or sources of orgasms. Future research could continue to explore the potential specificity of benefits from various types of orgasms from various forms of sexual activity. However, given the critiques of the early work in this area, future studies should seek to use more rigorous methods (e.g., more comprehensively assessing sexual health), seek more diverse populations (e.g., expanding to include sexual and gender minorities), and ground those studies within larger conceptual frameworks.

**Studying Sexual Health as a Dyadic Relationship Process.** Most of the records reviewed examined sexual health within individuals, collecting data from just one individual from each relationship for the individuals in relationships. When records do include partner data, analyses are typically examined for actor (i.e., within-person) effects, rather than examining how one partner’s experiences may affect the other partner’s experiences (i.e., partner effects). This primarily conceptualizes sexual health as a predominantly individual experience. However, as sexual health dimensions are often part of sexual activity with another person, and much of that sexual activity occurs within the context of romantic relationships, it is reasonable to propose that having satisfying and painless sex or orgasms (or unsatisfying or painful sex without orgasms) within a sexual coupling could very likely have a meaningful impact on the romantic or sexual partner in that couple. In fact, sexual activity with another person can be an emotionally charged experience (e.g., [135]; [318]), laden with expectations from both partners (e.g., [22]) and offering the possibility of intense intimacy (e.g., [156]). Any failures to meet those expectations or achieve fulfilling pleasure could therefore be troubling to one or both partners in that sexual coupling. Thus, by conceptualizing sexual health as a dyadic relationship process, relationship research could capture the dynamics of what can be an extremely intense experience within models of relationship functioning. It will also allow researchers to explore links between sexual health and other relationship processes, such as support, conflict, or even intimate partner violence.

One of the meta-analyzed records moved the examination of orgasms closer to a dyadic level by demonstrating links between *simultaneous* orgasms and relationship quality for both men and women ([51]). Similarly, another recent survey of 38,747 heterosexual men and women in 3+ year relationships from the United States shifted the focus toward the dyadic nature of orgasms by showing positive ties between an individual’s reports of their partner’s orgasm consistency and that individual’s own sexual satisfaction ([146]). Notably, those pseudo-partner effects (pseudo, as they are still only reported by a single individual) remained significant even after controlling for sexual activity and sexual communication. Although both of those records take important steps toward recognizing the dyadic nature of orgasms during sexual activity with a sexual/romantic partner, they were both limited by collecting data from only one individual in each relationship.

Within the current systematic review, only a small fraction of the records examined the links between sexual health dimensions and relationship functions by specifically collecting data from both romantic partners (e.g., [146]; [156]; [168]; [192]; [212]). Even fewer of these (e.g., [156]; [192]; [212]) analyzed the partner data dyadically (i.e., actor-partner interdependence modeling, APIM; e.g., [90]). For example, analyses in a sample of 128 Israeli heterosexual couples, [156] ([156]) demonstrated that individuals’ orgasmic consistencies were linked to higher sexual satisfaction (and lower attachment insecurities) for those individuals and their partners. Future work could advance our understanding of the salience of sexual health in the lives of couples by taking a similarly dyadic approach to studying it.

**Investigating Sexual Health as a Developmental Process.** Given the cross-sectional and largely incidental nature of the research linking sexual health to well-being, this work has been fairly atheoretical in its approach, often focusing on practical questions within clinical populations (e.g., to what extent is chronic pelvic pain linked to depressive symptoms and reduced orgasms; [21]) rather than developing testable theories. However, given the central role that sexual activity and sexual health can play in peoples’ lives and in their relationships, the current findings begin to suggest that various aspects of sexual health (i.e., learning about and growing comfortable with our bodies, embracing our sexual desires, exploring intimate activity, learning to communicate sexual needs, and developing the ability to have orgasms) could be conceptualized as a fundamental skills to be obtained as individuals become sexually active. Many records included in the current meta-analysis have measured orgasms as an ability/inability to have orgasms (e.g., [25]; [99]; [170]; [395]). Records like these have identified a population of women in their 40s, 50s, and 60s who have never experienced orgasms despite engaging in sexual activity (e.g., [132]; [395]), illustrating that some women have not and may never experience orgasms. Although biological issues might prevent some women from being able to experience orgasms or even pleasurable sex in general, for many women, sexual health difficulties could simply arise from a lack of knowledge of and comfort with their own bodies, and the associated skills needed to embrace sexual desire, communicate sexual needs, or achieve orgasms (either alone or during sexual activity with a partner). Thus, while many women are able to quickly learn those skills across their early sexual experiences, for many others, these skills might take years or decades after a sexual debut to acquire, delaying their abilities to achieve consistent, reliable, and satisfying sexual pleasure. Given the findings that childhood and adolescent sexual education can influence the achievement of these skills (e.g., [51]), it is possible that sexual health functioning could be considered a developmental process beginning in adolescence and extending through young adulthood and beyond. This process would not only include the physical development and sexual maturation of individuals’ bodies, but also the cognitive and emotional development of those individuals as they develop and embrace their own sexual identities, grow to know their own bodies, develop comfort and understanding of their own sexual interests and needs, establish their own sexual attitudes, and develop schemas and scripts for how sex fits into their lives and into their relationships. Although researchers have briefly commented on these ideas in previous work (e.g., [78]; [158]; [220]), this remains a largely unexamined area of research on sexual health, and especially orgasms. We would posit that these developmental factors will influence the quality of an individual’s sexual health both within and outside of committed relationships, most likely serving as enduring vulnerabilities and adaptive processes within the EVSA model. Thus, future work could take a more developmental and holistic perspective by assessing these various developmental processes over time appropriate developmental timeframes (e.g., during those formative years in adolescence) to place an individual’s current experience of sexual health functioning within a larger socio-emotional developmental context. Models based on such a developmental approach would likely offer novel insights to individual functioning across the lifespan.

**Linking Sexual and Physical Health.** The comprehensive review uncovered 9 previous studies linking orgasms to improved physical health. For example, analyses in a sample of 117 Turkish women linked sexual satisfaction, desire, lack of pain, vaginal lubrication, and consistency of orgasms to greater physical health on the SF-36 ([15]). Similarly, analyses in 76 Dutch women with a history of vulvar cancer and radiotherapy, the same sexual health dimensions were linked to greater physical health ([175]). Another study has linked men’s sexual intercourse frequency to greater longevity ([282]). Extending these studies, analyses in 143 Scottish men and women demonstrated a link between orgasms and greater resting heart rate variability, an indicator which is indicative of not only better mental health and emotion regulation, but also greater physical health and longevity ([95]), thereby highlighting a possible mechanism for the current findings. Although the path models tested in the current review conceptualized sexual health predicting physical health, it is likely that those forms of health are reciprocally related. In fact, in the case of chronic illness, it is more likely that physical health might function as the causal factor reducing the quality of sexual health and functioning over time. Future work could therefore extend the current findings by examining links between physical and sexual health in multi-wave longitudinal studies, thereby allowing those reciprocal directions of causality to be modeled. Future work could further extend the current body of work by examining the links between sexual health dimensions and other more concrete daily health outcomes, such as the number of colds, visits to a physician, and missed days at work, thereby extending these findings to nonclinical populations.

**Embracing Relationship and Sexual Orientation Diversity.** As most records in the current meta-analysis have only looked at heterosexual respondents, and presume monogamous relationship structures, it is difficult to ascertain whether these findings are applicable to sexual minorities or to those in non-monogamous relationships. Although each of sexual health components are likely to be important for individuals of all sexual orientations and in all forms of relationships, we would posit that it is also possible that these findings may vary across those groups. Given findings highlighting possible differences in the frequencies of sexual activity and orgasms across gay and lesbian relationships (e.g., [147]; [334]), it is also possible that the salience of sexual activity and orgasms might take on unique meanings within specific populations. Consistent with this, another body of work on relationship diversity has explored the characteristics of individuals in fundamental classes of monogamous and nonmonogamous relationships ([171]). Findings in a diverse sample of 1658 adult men and women suggested that although certain forms of nonmonogamy demonstrated comparable levels of individual and relationship functioning to monogamous relationships, the individuals in nonmonogamous relationships reported markedly different sexual attitudes from individuals in more traditional monogamous relationships, reporting higher socio-sexual orientations (i.e., comfort with and interest in casual sex) and higher sexual sensation seeking. Thus, it is likely that each of the sexual health components might take on different salience in nonmonogamous relationships. To extend the current findings, future work would therefore benefit from seeking greater diversity in the populations sampled and examining the correlates of sexual health components in non-heterosexual individuals and within nonmonogamous relationships.

**Examining Correlates of Exaggerating Sexual Pleasure.** The sexual health gender gap continues to be well documented in the United States (e.g., [14]; [32]; [147]; [306]; [335]), suggesting that women generally experience orgasms and sexual desire at notably lower rates than men, as well as sexual pain at higher rates than men. Given the intense expectations that can surround partnered sexual activity, orgasm difficulties (particularly those in women) likely exert pressure on individuals to exaggerate their own pleasure or even fake their orgasms. This is most commonly done in an effort to protect the feelings of a sexual partner (e.g., [134]; [267]; [319]), especially since greater amounts of sexual activity has been linked to lower relationship satisfaction for men whose female partners orgasm at low rates ([267]). The phenomenon of exaggerating pleasure fell beyond the scope of the current meta-analysis, but it remains a closely related process to orgasm difficulties and a growing body of studies have investigated this phenomenon. Although both men and women have reported having exaggerated pleasure or faked orgasms during intercourse, this appears to be much more prevalent in women than in men (e.g., [267]). In one of the first published records to link faking orgasms to relationship functioning, [131] ([131]) found that greater frequency of women’s faking orgasms was linked to lower relationship satisfaction, fewer self-reported orgasms, and greater reports of past infidelity in current relationships, highlighting the potential risks of faking orgasms. Extending this work, a number of labs have developed scales to assess common reasons for faking orgasms (e.g., [91]; [162]; [324]), as well as examining predictors of faking orgasms (e.g., [195]; [261]), and the possible impacts of faking orgasms on romantic relationships (e.g., [110]). As the current meta-analytic findings highlight the correlates of orgasms across multiple domains of functioning, the current findings could be extended meaningfully in future studies by examining the associated phenomenon of faking orgasms and the motives underlying such behavior.

## 5. Conclusions

The current meta-analytic review was the first of its kind to quantitatively integrate 49 years of research examining the correlates of sexual health across a wide range of studies. As the vast majority of that work was pragmatic rather than theoretically driven, the current review also sought to develop conceptual frameworks to theoretically integrate disparate lines of research into more comprehensive models of individual and relationship functioning. The meta-analytic results demonstrated links between sexual health components and physical, emotional, and relationship health and well-being, laying a foundation of support for the proposed models. The review also revealed a number of promising directions for future research: (1) examining mechanisms linking sexual health to individual and relationship functioning, (2) examining sexual health components as possible mediators and moderators with the EVSA and ASA models, (3) integrating predictors of sexual health as enduring traits within the EVSA model, (4) examining possible directions of casual influence using multi-wave longitudinal studies, (5) examining the specificity of various types and sources of orgasms, (6) collecting dyadic data to fully model sexual pleasure as a dyadic interpersonal process, (7) modeling sexual health and more specifically the ability to achieve orgasms as developmental processes involving a discrete set of skills and stages, (8) deepening our understanding of links between sexual and physical health, (9) embracing diversity in relationship commitment structures and sexual orientations, and (10) extending work on exaggerating pleasure or faking orgasms to hide orgasmic difficulties from a romantic or sexual partner. Thus, the current meta-analytic review not only synthesizes quantitative findings but also offers concrete guidelines for extending the past 49 years of research on the correlates of sexual health in a theoretically grounded manner.

## Figures and Tables

**Figure 1 behavsci-15-01636-f001:**
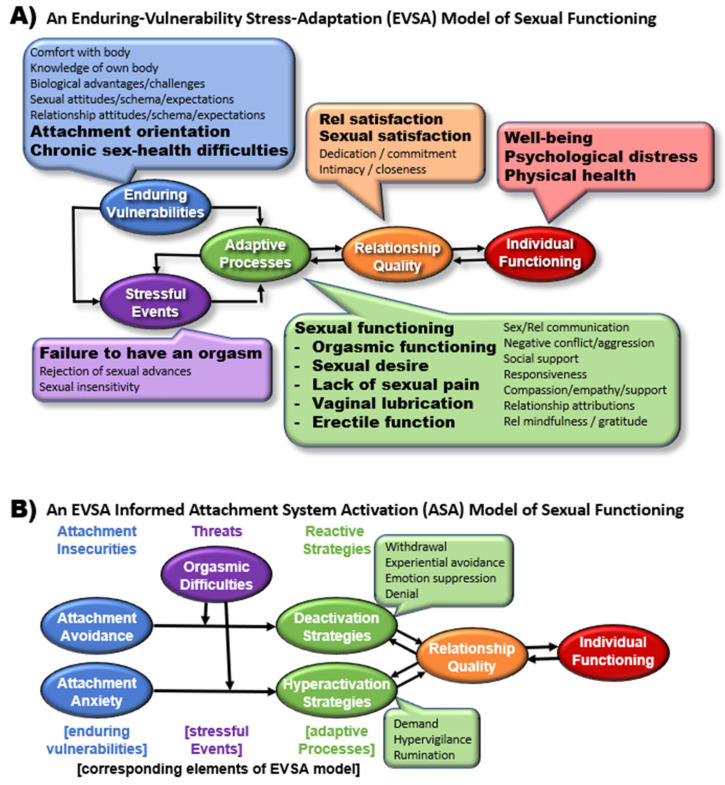
Conceptual Models Integrating Sexual Health into Models of Relationship and Individual Functioning. Note. Bolded constructs in panel A correspond to search terms used to identify potential correlates.

**Figure 2 behavsci-15-01636-f002:**
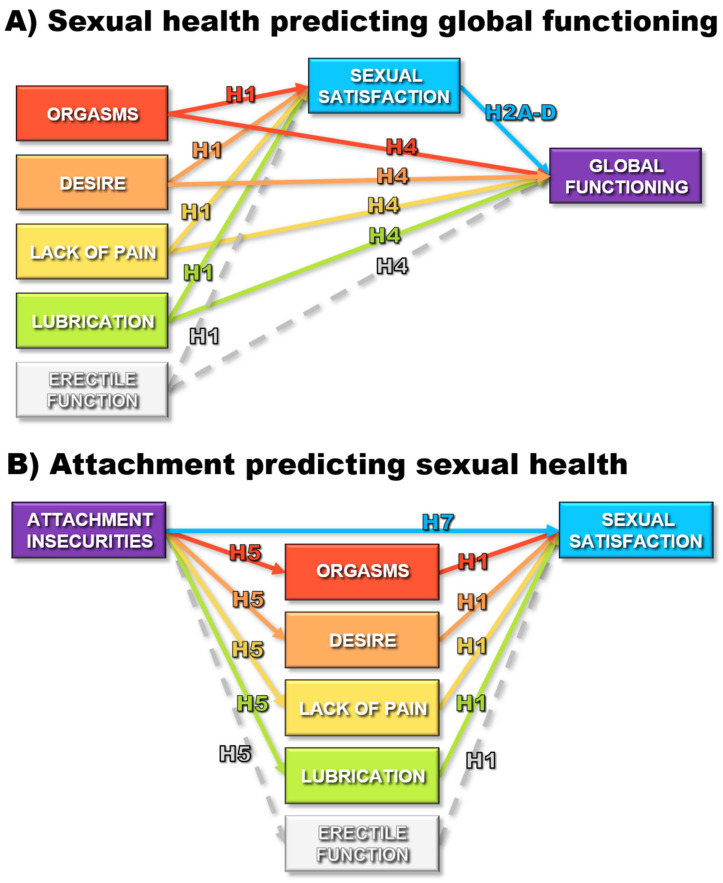
Hypotheses to be Tested in Path Models. Note: The arrows to and from Erectile Function have been dashed as there was an insufficient number of studies to test them with meta-analysis.

**Figure 3 behavsci-15-01636-f003:**
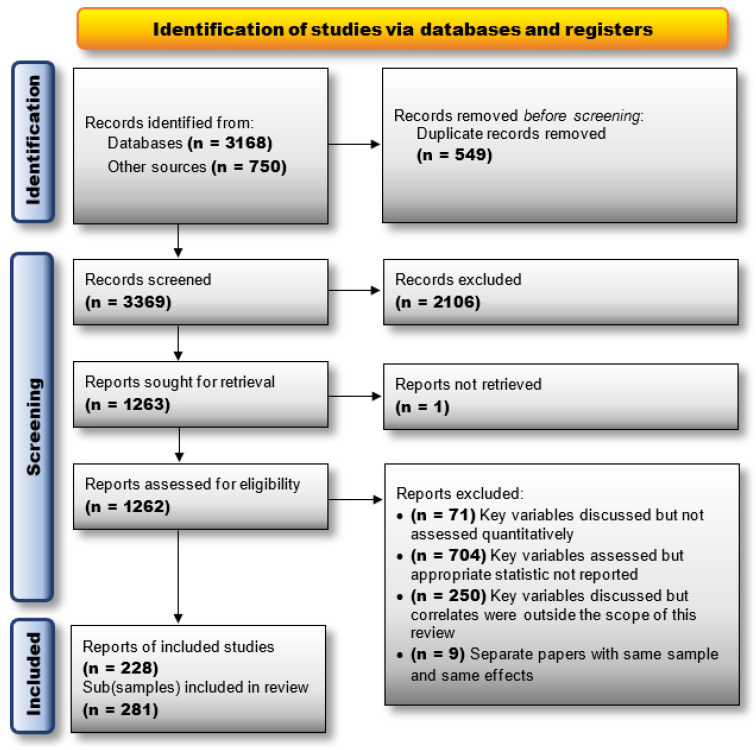
Flowchart detailing record search, record screening, data inclusions, and data exclusions. Note. The PsycINFO search was set up to automatically remove duplicate records.

**Figure 4 behavsci-15-01636-f004:**
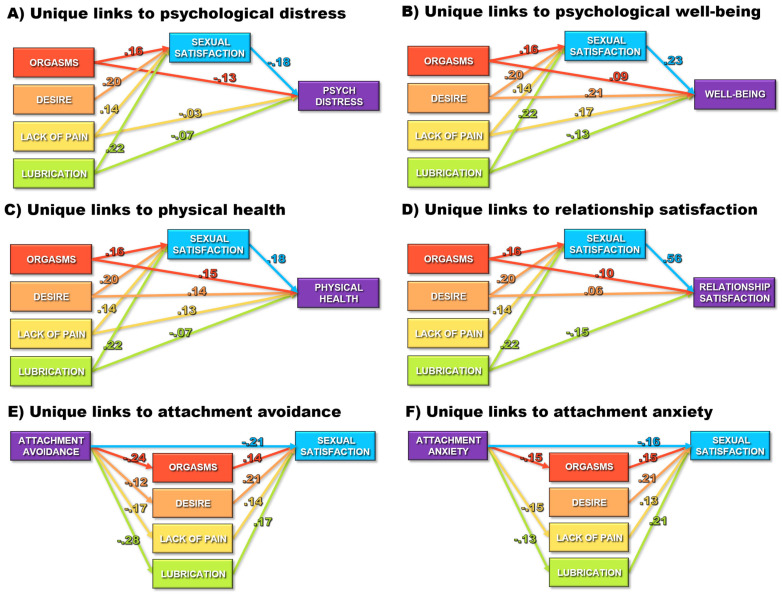
Significant Results of Path Models Examining the Unique Predictive Links of the Dimensions of Sexual Health. Note. Only paths significant at *p* < .01 are shown.

**Table 1 behavsci-15-01636-t001:** Summary of (Sub)samples in Analysis.

Category		# Subsamples	Category	# Subsamples	Category/Code	# Subsamples
Subset (# Reporting)		(# of Effects)	Subset			
Summary			Clinical population ^2^		Country	
Distinct samples		281	Addison’s disease	1	Australia	4
Unique subjects		248,021	Ankylosing spondylitis	2	Belgium	2
Distinct effects		1201	Brain or spinal cord injury	4	Brazil	9
Publication type			Cancer	7	Bulgaria	1
Peer-reviewed article		260	Chronic pain	4	Canada	27
Thesis/dissertation		21	Depression &/or anxiety	9	Chile	2
Publication year			Dialysis	2	China/Hong Kong	10
1971–1990		7	Fertility treatment	5	Czech Republic	5
1991–2000		10	Gynecological disorders	11	Denmark	2
2001–2010		47	Heart disease/hypertension	4	Egypt	1
2011–2020 (February)		141	Heroin use	1	Finland	5
2020 (February) to 2025 (October)		76	Hyperthyroidism	1	Germany	4
Sex Health ^1^			Incontinence	1	Greece	1
	Sexual Satisfaction	(471)	Lupus	1	Hungary	1
Orgasm		(474)	Metabolic syndrome	1	India	1
Desire		(358)	Migraines	1	International	2
Lack of pain		(184)	Multiple sclerosis	4	Iran	9
Lubrication		(195)	Nonspecific medical ^3^	2	Israel	5
Erectile function		(51)	Obesity &/or eating disorder	3	Italy	12
Correlate			Parkinson’s disease	1	Japan	1
Psychological distress		(329)	Pre-, post-, &/or menopausal	8	Malaysia	3
Psychological well-being		(85)	Pregnant &/or postpartum	9	Mexico	1
Physical health		(58)	PTSD	1	Morocco	1
Attachment anxiety		(41)	Renal transplant	1	Netherlands	5
Attachment avoidance		(41)	Rheumatoid arthritis	1	Norway	2
Relationship satisfaction		(169)	Schizophrenia	2	Poland	7
Type of population ^2^			Sexual abuse	2	Portugal	12
College students		16	Sexual dysfunction	12	Scotland	1
Community		145	Sjogren’s syndrome	1	South Korea	1
Clinical		120	Sleep apnea	2	Spain	4
Gender			Stroke	1	Sweden	3
Female only study		183	Vitamin D3 deficiency	1	Switzerland	3
Male only study		60	Other medical	6	Taiwan	2
Both genders		38	Average age (*k* = 199)		Turkey	21
Weighted percent female		75%	Lowest	14.6	UK	5
Relationship length (*k* = 41)			Highest	73.5	US	88
Lowest		1.1	Weighted mean	37.1	Other	18
Highest		34.1	In a relationship (*k* = 211)		Caucasian	
	Weighted mean	8.0	Weighted %	77.1	Weighted %	63.80%

Note. The 281 unique subsamples presented in this table were drawn from the 228 unique records presented in the following table. ^1^ These numbers represent the number of subsamples providing at least one effect involving the construct listed. ^2^ These numbers sum to more than 205 as these are not mutually exclusive categories (e.g., some samples were made up of both students and people from the community). ^3^ These two studies were not focused on a specific clinical population but instead included samples of elderly individuals with a wide range of medical diagnoses.

**Table 2 behavsci-15-01636-t002:** Characteristics of Individual Study Subsamples in Analysis.

Author (Year)	Record Type	N	Country	Subsample	Mean Age	Percent White	Sex Health Dimension(s)	Sex Health Scale(s)	Correlate Dimension	Correlate Scale	# Effects
[1] ([1])	PR	38	Egypt	Muslim males w/lifelong delayed ejaculation	30.8	0	O	IIEF	depressive sxs	SRADM	1
[2] ([2])	PR	228	Iran	Women w/& w/out abnormal sexual function	30.4	0	SODPL	FSFI	perceived stress	PSS	5
[5] ([5])	PR	65	Belgium	Women aged 18–58 w/orgasm difficulties	32.7	100	SO	FSFI	only sex health *		1
[6] ([6])	PR	84	Belgium	Women w/endometrial cancer	63.0	100	O	SSFS	relationship sat	DAS	1
[7] ([7])	PR	204	Italy	Women w/& w/out metabolic syndrome	40.0	100	O	FSFI	depressive sxs	MHQ	2
[13] ([13])	PR	1997	US	Middle-aged women w/multiple medical diagnoses	60.2	36	SODPL	1 item	depression diagnosis	medical diagnosis	5
[15] ([15])	PR	177	Turkey	Women in sexual relationships w/& w/o sexual dysfunction	32.3	0	SODPL	FSFI	mental health, vitality, physical health	SF-36	15
[16] ([16])	PR	149	Germany	Pregnant women w/& w/o depression	28.9	100	SODL	MGH	depression diagnosis	CIDI	4
[17] ([17])	PR	110	Greece	Healthy women & obese women preparing for bariatric surgery	34.4	100	SODPL	FSFI	depressive sxs	BDI	5
[19] ([19])	PR	80	Turkey	Women w/& w/o hyperthyroidism	36.8	0	SODPL	FSFI	depressive sxs	BDI	5
[21] ([21])	PR	72	US	Men w/chronic pelvic pain syndrome	40.8	66	O	BSFQ	depressive sxs	CESD	1
[23] ([23])	PR	236	Iran	Women in fertility treatment	26.4	0	SOD	DSM-IV	relationship sat	1 item	4
[24] ([24])	PR	269	Italy	Women who have given birth 6 mo. ago	34.4	100	SO	FSFI	only sex health *		1
[25] ([25])	PR	4339	Chile	Women & men aged 18–69	39.7	0	SO	1 item	only sex health *		2
[26] ([26])	PR	87	Italy	Menstruating women	28.7	100	O	1 item	depressive sxs	BDI	1
[27] ([27])	PR	120	US	Women in heterosexual & lesbian relationships	29.4	77	SODPL	FSFI	only sex health *		16
[30] ([30])	PR	149	Portugal	Adult women	25.0	0	SO	FSFI	only sex health *		1
[32] ([32])	PR	750	Canada	Women & men in same or opposite sex relationships	30.0	0	SO	IIEF	only sex health *		2
[35] ([35])	PR	627	Bulgaria	Middle-aged pre- & post-menopausal women w/& w/o hormone replacement therapy	48.0	100	O	1 item	coping with life	1 item	1
[52] ([52])	PR	1570	Czech Republic	Women & men aged 35–65 in relationships	48.8	100	SO	1 item	life sat, relationship sat	LiSat	6
[55] ([55])	PR	58	US	Middle-aged women	47.6	0	O	1 item	Depressive sx, positive affect	1 item, 7 items	2
[57] ([57])	PR	507	UK	Pre- & post-menopausal twin women	56.3	100	SODPL	FSFI	relationship sat	1 item	14
[58] ([58])	PR	866	UK	Monozygotic twin women discordant for sexual dysfunction	55.0	100	SODPL	FSFI	relationship sat	1 item	5
[62] ([62])	PR	518	Italy	Sexually active women aged 40–55 at gynecological clinics	49.4	100	SO	FSFI	only sex health *		1
[64] ([64])	PR	108	Turkey	Women aged 22–51 w/vitamin D3 deficiency	34.9	0	SODPL	FSFI	depressive sx	BDI	5
[65] ([65])	MT	145	US	Adult women	25.8	0	SODPL	FSFI	only sex health *		10
[67] ([67])	PR	157	Spain	College women aged 20–45	22.8	0	SO	3 items	mental health	WHO-5	3
[70] ([70])	PR	309	Italy	Obese & not obese women w/& w/o Binge ED	39.1	100	SODPL	FSFI	depressive sx	BDI	15
[72] ([72])	PR	102	Turkey	Married women in treatment for hypertension	55.1	0	SOD	ASEX	relationship sat	MAT	3
[75] ([75])	PR	555	Taiwan	Pregnant women	33.0	0	SODPL	FSFI	depressive sx	CESD	5
[76] ([76])	PR	1026	Taiwan	Women aged 40–65	48.5	0	SODPL	FSFI	mental health, physical health	SF-12, SF-12	10
[79] ([79])	PR	160	China	Sexually active men aged 50+ from a primary care treatment center	55.0	0	SOE	IIEF	depressive sx	GDS	3
[80] ([80])	PR	57	Canada	Post-partum women w/& w/o depression	34.3	0	SODPL	FSFI	depressive sxs	EPDS	5
[84] ([84])	PR	323	UK	Adult women	24.4	70	O	4 items	attachment	ECR	2
[88] ([88])	MT	158	US	Women w/gynecologic or breast cancer	57.7	92	SO	DSFI; 1 item	depressive sxs, relationship sat	CESD, single item?	3
[89] ([89])	DD	531	US	Adult women	27.0	78	SO	ISS, 1 item	psychological sxs, attachment, relationship sat	BSI, ECR, RAS	7
[93] ([93])	PR	30	Portugal	Women currently in sexual relationships	25.8	0	SO	1 item	relationship sat	PRQC	2
[94] ([94])	PR	70	Scotland	College sexually experienced women	25.7	100	O	1 item	attachment	ECR	2
[96] ([96])	PR	116	Portugal	Adult women & men	24.9	0	SO	1 item	only sex health *		2
[98] ([98])	PR	16	Malaysia	Women aged 28–48 w/PCOS	33.4	0	SODPL	FSFI	depressive sxs	DASS-21	5
[99] ([99])	PR	206	US	Married undergrad & graduate women	28.1	0	SO	1 item	only sex health *		1
[101] ([101])	DD	220	US & Canada	Couples w/o current sexual, medical, or psychiatric diagnoses or treatments	26.5	86	SO	2 items	only sex health *		1
[102] ([102])	PR	96	Brazil	Women attending routine gynecology checkup	38.5	0	SOD	FSQ	anxiety sxs	BAI	3
[104] ([104])	PR	46	Brazil	Men aged 18–60 w/spinal cord injury	34.0	0	SO	IIEF	only sex health *		2
[106] ([106])	PR	23	Turkey	Women w/ankylosing spondylitis	39.3	0	SODPL	FSFI	depressive sxs, vitality, physical health	BDI, SF-36, SF-36	15
[107] ([107])	PR	200	US	Undergrad women & men from U of California, Santa Barbara	19.6	59	O	1 item	relationship sat	MOQ	1
[108] ([108])	PR	206	US	Female & male college students	19.3	68	O	1 item	relationship sat	MOQ	2
[113] ([113])	PR	130	Italy	Women w/& w/out anxiety diagnoses	29.5	100	SODPL	FSFI	anxiety sxs	STAI	10
[117] ([117])	PR	57	Turkey	Women heroin users	26.2	0	SODPL	FSFI	depressive sxs	BDI	7
[118] ([118])	PR	142	Turkey	Women w/& w/o migraines	34.0	0	OL	FSFI	depressive sxs	BDI	2
[122] ([122])	PR	19	Netherlands	Men w/history of strokes	58.8	100	SODE	IIEF	depressive sxs	SCL	4
[123] ([123])	PR	576	Canada	Undergrad women from U of British Columbia	20.8	37	SODPL	FSFI	attachment	ECR	10
[124] ([124])	PR	613	UK	Women aged 18–75	50.0	0	OL	1 item, 1 item	depressive sxs, marital difficulties	HAD, 1 item	4
[126] ([126])	PR	279	US	Sexual and gender minority individuals assigned female at birth who had been in sexual or romantic relationships within the last 6 mo.	21.2	19	SO	3 items, 2 items	relationship sat	RAS	3
[127] ([127])	PR	70	US	Women w/gynecologic cancer	53.0	80	OP	WSIDI	depressive sxs	PHQ	2
[129] ([129])	PR	86	US	Men & women who had or had not learned & practiced the coital alignment technique	41.1	0	SO	1 item	only sex health *		1
[130] ([130])	MT	330	International	Portuguese or English speaking, sexually active women & men	28.4	0	SOD	1 item	only sex health *		4
[131] ([131])	PR	112	US	Female undergraduate students in relationships	18.8	0	O	1 item	relationship sat	RAS	1
[133] ([133])	PR	281	US	Women w/depression	38.3	88	OD	CSFQ	depressive sxs	HAMD	2
[132] ([132])	PR	326	US	Men w/major or atypical depression	39.5	80	OD	DISF	depressive sxs	HAMD	2
[136] ([136])	PR	64	US	Women & men w/schizophrenia in outpatient treatment	43.5	70	SOD	CSFQ	depressive sxs, quality of life	HAMD, QOL	12
[143] ([143])	PR	336	Australia	Sexually active women aged 19–65	29.5	100	SO	FSFI	depressive sxs,	DASS	2
[146] ([146])	PR	38747	US	Heterosexual women & men in a relationship for at least 3 years	39.4	87	SO	1 item	only sex health *		4
[151] ([151])	PR	54	US	Heterosexual undergrad women in sexual relationships	19.0	0	SO	1 item	only sex health *		1
[163] ([163])	PR	311	US	Women enrolled in the Penn Ovarian Aging Study	48.0	51	SODL	FSFI	anxiety sxs	ZAI	4
[155] ([155])	PR	256	Israel	Heterosexual couples (dyads)	34.4	87	SOD	ISS, ISBI	only sex health *		6
[156] ([156])	PR	256	Israel	Married or cohabitating couples, together 3+ yrs & co-parenting 1+ children (dyads)	34.4	0	SOD	ISS, 3 items	attachment	ECR	12
[160] ([160])	PR	1363	Portugal	Sexually active men w/& w/o erectile dysfunction	37.7	0	SODE	IIEF	only sex health *		6
[161] ([161])	DD	1576	US	Adult women & men collected online	32.0	79	O	1 item	attachment, relationship sat	ECR, PRQC	3
[165] ([165])	PR	12	Italy	Men aged 21–59 w/Addison’s disease	40.5	100	SOD	IIEF	only sex health *		2
[166] ([166])	DD	430	US	Heterosexual undergrad women w/a sex partner in the last year	19.9	38	SO	MSSCQ-S, FSFI	only sex health *		2
[167] ([167])	PR	2173	Finland	Twin women	25.5	100	SODPL	FSFI	only sex health *		10
[168] ([168])	PR	13498	China	Newlywed women & men in Shanghai (dyads)	27.8	0	SO	1 item	relationship sat	1 item	4
[169] ([169])	PR	159	Switzerland	Adult women	31.8	100	SODPL	FSFI	relationship sat,	RAS	15
[170] ([170])	PR	2250	Finland	Women & men aged 18–74	42.2	100	SO	2 items	only sex health *		2
[173] ([173])	PR	1207	US	Undergrad sexually active men using & not using ED meds	21.9	73	SODE	IIEF	only sex health *		6
[174] ([174])	PR	354	Turkey	Men w/erectile dysfunction	37.9	0	SO	IIEF	only sex health *		1
[175] ([175])	PR	76	Netherlands	Women w/history of vulvar cancer & radiotherapy	68.0	100	SOD	FSFI	physical health	SF-36	3
[179] ([179])	PR	508	Hungary	Undergrad women & patients w/endometriosis or PCOS	27.9	100	SODPL	FSFI	only sex health *		10
[178] ([178])	PR	1843	US & Hungary	Adult women	28.8	0	SO	1 item	relationship sat	1 item	2
[181] ([181])	DD	99	US	Women in relationships	27.5	97	SO	SII 1 item	only sex health *		1
[182] ([182])	PR	451	Germany	CBT outpatients w/anxiety or depression diagnosis	36.0	100	SODLE	MGH	only sex health *		9
[185] ([185])	PR	98	US	Fort Hood, TX military wives	26.7	68	SOD	ISS, 1 item	only sex health *		3
[187] ([187])	PR	32	South Korea	Women w/& w/o stress urinary incontinence	41.7	0	SO	FSFI	only sex health *		1
[189] ([189])	PR	75	Denmark	Gynecological cancer patients (cervical, endometrial, ovarian) aged 28–77	58.0	100	SOL	SVQ	only sex health *		2
[192] ([192])	PR	284	US	Heterosexual couples (dyads)	32.4	90	SO	NSSS-S, 1 item	only sex health *		2
[194] ([194])	DD	104	US	Married women aged 25–40	31.7	0	SO	ISS, 1 item	only sex health *		1
[199] ([199])	PR	136	US	Adult women	20.7	82	OP	FSDS-R	depressive sxs	CESD	2
[197] ([197])	PR	1258	US	Undergrad women & men from Kent State U w/depression &/or anxiety	19.6	0	SODPLE	FSFI/MSFI, PFSF	general distress	MASQ	3
[196] ([196])	PR	171	US	College women free of antidepressants	20.1	81	ODL	FSFI	depressive sxs	CESD	22
[198] ([198])	PR	150	US	Postmenopausal women w/chronic insomnia	56.4	52	SODPL	FSFI	depressive sxs	BDI	5
[201] ([201])	PR	33	Turkey	Women getting hemodialysis	41.0	0	SODPL	FSFI	depressive sxs	BDI	5
[205] ([205])	PR	501	US	Undergrad women from Ohio U w/& w/o history of sexual assault	18.9	88	ODPL	FSFI	depressive sxs	TSC-D	10
[206] ([206])	PR	225	Turkey	Dialysis patients	43.0	0	SOE	GRISS	depressive sxs	BSI	3
[208] ([208])	PR	295	Australia	Postpartum women	29.7	0	SO	FSFI	depressive sxs, relationship sat	PHQ, RAS	3
[210] ([210])	PR	60	Morocco	Women w/rheumatoid arthritis	45.2	0	SODPL	FSFI	depressive sxs, vitality, physical health	HADS-D, SF-36, SF-36	15
[212] ([212])	PR	170	Czech Republic	Heterosexual couples (dyads)	27.1	100	SO	1 item	relationship sat	DAS	4
[214] ([214])	PR	200	Israel	Women aged 21–44 attending antenatal classes	30.5	0	SODPL	FSFI	relationship sat	EMS	10
[215] ([215])	PR	84	Sweden	Women & men post traumatic brain injury	40.0	100	SO	SIS, 1 item	only sex health *		4
[227] ([227])	PR	299	Switzerland	Adult women	50.0	100	O	1 item	depressive sxs	SCL	1
[229] ([229])	PR	3366	US	Heterosexual newlywed couples (dyads)	28.6	62	SO	GRISS, 1 item	relationship sat	CSI	4
[234] ([234])	PR	204	Poland	Women & men w/multiple sclerosis	50.3	100	SODPLE	SFQ	depressive sxs	BDI	9
[238] ([238])	PR	117	Spain	Post-menopausal women	57.0	0	OD	CSFQ	depressive sxs	CESD	2
[242] ([242])	DD	243	US	Undergrad women from Oklahoma State U	19.6	69	SODP	GSF	only sex health *		6
[243] ([243])	DD	92	US	Men enrolled in Osher Lifelong Learning Institute at U of Nevada, Las Vegas	73.5	89	SOE	MSHQ	perceived stress	PSSI	3
[249] ([249])	PR	52	Tunisia	Women aged 27–57 w/& w/out multiple sclerosis	37.7	0	SODPL	FSFI	depressive sxs	BDI	5
[251] ([251])	PR	137	Canada	Women reporting pain during sex	32.2	0	SO	FSFI	only sex health *		1
[253] ([253])	DD	95	US	Arizona married female community college students	38.5	0	SO	SIS	relationship sat	MAT	3
[255] ([255])	PR	90	Brazil	Men w/spinal cord injury—retroactive pre-injury report	33.5	0	SODE	1 item	only sex health *		3
[256] ([256])	PR	93	Switzerland	Women aged 40–73	52.5	100	SODPL	FSFI	life sat, relationship sat	SWLS, RAS	10
[263] ([263])	PR	82	US	Women &men aged 29–74 in treatment for chronic pain	49.9	67	O	DISF	depressive sxs	CESD	1
[264] ([264])	PR	65	Spain	Women w/lupus	39.0	0	SODPL	FSFI	depressive sxs, vitality, physical health	SCL, SF-36, SF-36	15
[266] ([266])	PR	100	Iran	Sexually active married women w/& w/out recurrent vulvovaginal candidiasis	31.5	0	SODPL	FSFI	depressive sxs	DASS-21	5
[270] ([270])	PR	1200	Iran	Reproductive aged women	29.9	0	SO	FSFI	anxiety	---	2
[271] ([271])	PR	157	Brazil	Hypertensive women	56.4	0	SODPL	FSFI	depressive sxs	HADS-D	15
[272] ([272])	PR	321	Malaysia	Women w/& w/out severe delivery complications	30.2	0	SO	FSFI	only sex health *		1
[275] ([275])	PR	26	Turkey	Patients w/obstructive sleep apnea-hypopnea	45.8	0	SODPL	FSFI	depressive sxs	BDI	5
[277] ([277])	PR	85	Canada	Undergrad women in relationship	20.8	100	SODPL	FSFI	only sex health *		10
[279] ([279])	PR	89	Turkey	Adults w/Parkinson’s disease	67.7	0	ODLE	ASEX	depressive sxs	HAMD	4
[280] ([280])	PR	98	Turkey	Women & men w/renal transplants	36.1	0	OD	ASEX	depressive sxs	BDI	3
[281] ([281])	PR	131	Iran	Peri-menopausal women at a health center	55.8	0	SODPL	FSFI	stress, relationship sat	---	20
[283] ([283])	PR	44	Italy	College women aged 21–47	26.1	100	SODPL	FSFI	depressive sxs	SCL	5
[284] ([284])	PR	2411	China	Sexually active women & men aged 20–64 in long-term relationships	40.4	0	SO	1 item	only sex health *		2
[286] ([286])	PR	248	Portugal	Women & men in heterosexual couples	29.5	0	SODLE	GMSEX, FSFI	only sex health *		12
[287] ([287])	PR	36	Brazil	Postmenopausal women aged 45–65 at a climacteric clinic	55.4	0	SODPL	FSFI	mental health, vitality, physical health	SF-36	15
[292] ([292])	PR	102	Germany	Adult women	36.7	100	SO	4 items, 1 item	only sex health *		1
[294] ([294])	PR	275	US	College aged sexually active heterosexual women	20.5	97	SO	PSSI, 1 item	only sex health *		1
[295] ([295])	MT	248	US	Male combat veterans w/PTSD aged over 18	---	85	SODE	IIEF	attachment	ECR	19
[297] ([297])	DD	835	US	Undergrad sexually active women from U of North Texas	21.2	60	O	1 item	depressive sxs	QIDS	1
[302] ([302])	DD	63	US	Women w/endometriosis & suffering from chronic pelvic pain	33.3	87	OD	DISF	depressive sxs	CESD	3
[310] ([310])	PR	259	US	Women w/& w/o sexual arousal disorder	40.1	76	SODPL	FSFI	only sex health *		20
[308] ([308])	PR	193	Canada	Women w/sexual arousal disorder & their male partners (dyads)	31.2	86	SO	FSFI	depressive sxs, relationship sat	BDI, CSI	10
[313] ([313])	PR	574	Denmark	Heterosexual, sexually active couples dealing w/breast cancer (dyads)	55.8	100	SO	PROMIS	only sex health *		2
[314] ([314])	PR	2068	US & Hungary	Adult women	29.1	0	O	1 item	relationship sat	1 item	1
[315] ([315])	PR	2304	US & Hungary	Adult women	28.8	0	SO	1 item	depression diagnosis; relationship sat	1 item each	3
[317] ([317])	PR	159	Turkey	Sexually active women at routine gynecology visits	33.7	0	SODPL	FSFI	depressive sxs	BDI	15
[320] ([320])	PR	70	Turkey	Men aged 20–50 w/ankylosing spondylitis	36.4	0	SODE	IIEF	depressive sxs	HADS-D	4
[321] ([321])	PR	2087	Italy	Women at menopausal clinics	54.0	100	O	1 item	mental health, physical health	SF-12	2
[323] ([323])	PR	206	US	Women in lesbian couples	33.7	73	SO	1 item	only sex health *		1
[325] ([325])	PR	493	US	Adult women	31.0	0	O	1 item	mental adjustment	1 item	1
[328] ([328])	PR	31581	US	Women from a nationally representative sample	49.5	81	O	3 items	depressive sxs	CSFQ	1
[329] ([329])	PR	129	Poland	Men aged 31–76 w/& w/out obstructive sleep apnea	57.9	100	SO	IIEF	depressive sxs, relationship sat	HADS-D	5
[333] ([333])	PR	55	US	Women aged 41–69 surviving breast cancer	53.4	100	SOD	FSFI	relationship distress	MSI-R	5
[337] ([337])	PR	319	US	Women in sexually active heterosexual relationships w/& w/o sexual arousal disorder	22.8	63	SODPL	FSFI	relationship sat	DAS	28
[336] ([336])	PR	176	US	Women w/& w/out history of childhood sexual abuse	33.3	71	SODPL	SSS-W, FSFI	only sex health *		10
[340] ([340])	PR	289	UK	College women & internet users aged 17–69	23.5	73	SO	FSFI	only sex health *		1
[345] ([345])	PR	126	Japan	Women in partnered relationships	38.1	0	SODPL	FSFI	only sex health *		10
[346] ([346])	PR	158	China	Women & men industrial workers	40.2	0	SO	MSQ, 1 item	only sex health *		1
[347] ([347])	PR	926	Portugal	Sexually active heterosexual premenopausal women	25.2	0	O	1 item	depressive sxs, positive affect, relationship sat	PANAS	3
[348] ([348])	PR	116	India	Men w/& w/o depression	32.4	0	SODE	IIEF	depressive sxs	BDI	4
[349] ([349])	PR	255	Canada	Women w/sexual dysfunction	---	82	O	FSFI	depressive sxs, relationship sat	BDI, 1 item	2
[352] ([352])	PR	350	US	Lesbian women	35.5	86	SODPL	FSFI	psychological sxs, relationship sat	BSI, DAS	20
[353] ([353])	PR	42	Norway	Heart disease patients	58.8	0	SOD	1 item	only sex health *		3
[354] ([354])	PR	93	Netherlands	Testicular cancer patients	29.4	100	OE	IIEF	depressive sxs	CESD	2
[355] ([355])	PR	311	Canada	College women aged 17–38	19.2	81	ODPL	FSFI	depressive sxs	DASS	10
[357] ([357])	PR	94	Sweden	Pregnant women & their mothers		100	SO	1 item	only sex health *		1
[363] ([363])	PR	399	Netherlands	Undergrad women	21.7	100	SODPL	FSFI	attachment	ECR	13
[364] ([364])	PR	85	Netherlands	Women w/& w/out Sjogren’s syndrome	45.4	100	SO	FSFI	depressive sxs	HADS-D	2
[367] ([367])	PR	200	Mexico	Young adult women from Mexico City	26.3	0	O	FSFI	depression diagnosis	1 item	1
[371] ([371])	PR	2195	US	Women & men in national sample	36.8	87	SO	1 item	relationship sat	2 items	4
[373] ([373])	PR	15	US	Treatment of preorgasmic women	27.5	88	SO	1 item	only sex health *		1
[379] ([379])	PR	720	Czech Republic	Women in relationships aged 35–65	47.6	100	SO	1 item	life sat, relationship sat	1 item each	3
[385] ([385])	PR	2081	Finland	Women aged 33–43	37.5	100	SODPL	FSFI	only sex health *		10
[384] ([384])	PR	2081	Finland	Pregnant women, & women w/children	37.5	100	SOD	FSFI	relationship sat	PRQC	5
[386] ([386])	PR	1315	US	Women reporting committed or casual sex	23.7	0	SO	SSS-W, FSFI	only sex health *		2
[390] ([390])	PR	300	Turkey	Menopausal women	53.6	100	SODPL	FSFI	depressive sxs	BDI	5
[391] ([391])	PR	298	Turkey	Pregnant married women	27.5	0	SO	GRISS	relationship sat	MAS	3
[392] ([392])	PR	80	Brazil	Women w/premature ovarian insufficiency	38.4	0	SODPL	FSFI	mental health, physical health	WHOQoL	10
[393] ([393])	PR	269	Malaysia	Married heterosexual couples in fertility treatments (dyads)	33.2	0	SODPLE	FSFI	only sex health *		16
[394] ([394])	PR	797	US	Married women	---	0	SO	3 items, 1 item	relationship sat	1 item	3
[395] ([395])	PR	1518	China	Married women	38.9	0	SODPL	1 item	life sat, relationship sat	1 item each	11
[396] ([396])	PR	1789	China	Adult women	37.2	0	SO	1 item	only sex health *	1 item	1
**ARTICLES PUBLISHED SINCE FEB-2020**											
[236] ([236])	PR	148	US	Sexually active female undergraduate survivors of sexual violence	22.4	43	SOD	SSS-W, FSFI	only sex health *		3
[159] ([159])	PR	1036	Canada	Sexually active high school students from Quebec	14.6		SOD	GMSEX, ASEX	only sex health *		3
[191] ([191])	PR	158	US	Individuals in romantic relationships ≥ 6 mo.	24.7	75	SO	GMSEX, 1 item	only sex health *		1
[226] ([226])	PR	3314	US	Sexually active adults in rom. relationships ≥ 2 yr.	34.3	77	SO	GMSEX, 1 item	relationship sat	CSI-4	6
[3] ([3])	PR	460	Iran	Sexually active middle-aged empty-nester women	51.7	0	SODPL	LSS, FSFI	depressive sx	UCLA	20
[246] ([246])	PR	104 (208)	Spain & Latin Am countries	Same-sex Spanish-Hispanic-Latin couples together ≥ 3 mo.	28.0	0	SO	GMSEX, ORS	relationship sat	GMREL	6
[296] ([296])	PR	1241	US	Sexually active adults in rom. relationships	34.0	82	SD	QSI-6, 4-items	depressive sxs, relationship sat	PHQ-9, CSI-8	9
[109] ([109])	PR	319	US	University adults in sexual relationships	19.0	62	O	1 item	relationship sat	MOQ	1
[378] ([378])	PR	2028	Hong Kong	Heterosexual couples	32.0	0	SODPLE	LSS, FSFI	only sex health *		16
[59] ([59])	PR		US	Adults in committed relationships ≥ 2 yr.	---	78	SO	PN-RQ(mod), 1 item	relationship sat	PN-RQ	3
[82] ([82])	PR	112	Turkey	Women with breast cancer and healthy controls	52.5		SODPL	FSFI	depressive sx	HAMD	5
[222] ([222])	PR	2854	Canada	Québécois adults from community & sex clinic	38.1		S	GMSEX	depressive sxs, relationship sat	K-6, DAS-4	4
[362] ([362])	PR	63	Netherlands	Women undergoing breast reconstruction post-mastectomy	45.0		SL	FSFI	only sex health *		1
[383] ([383])	PR	81	Poland	Women with connective tissue diseases	33.7		SODPL	FSFI	depressive sxs, relationship sat, life sat, phys health	HADS-M, PERMA-Profiler	20
[10] ([10])	PR	185 (370)	Canada	Mixed-sex Québécois couples seeking reproductive services for infertility	32.7	92	SODPL	FSFI, IIEF	life sat, phys health, relationship sat	FertiQoL	24
[331] ([331])	PR	24	US	Men post-myocardial infarction	62.5	96	SODE	MSFI	only sex health *		6
[202] ([202])	PR	217	Turkey	Women who completed treatment for early-stage breast cancer	45.0		SODPL	FSFI	depressive sx	HADS-D	15
[149] ([149])	PR	1084	US	Men engaging in receptive anal sex in the last 6 mo.	32.0	70	SODE	ASFI	depressive sx	BSI	10
[241] ([241])	PR	155	Senegal	Sexually active women living in Dakar	34.0	0	SO	2 single items	only sex health *		1
[382] ([382])	PR	243	US	Adults who have hooked up in the last month via an app	26.8	80	SODPLE	FSFI, MSHQ	only sex health *		11
[157] ([157])	PR	91	Canada	Québécois cis-gender adults living with spinal cord injuries	40.8		SO	GMSEX, FSDS-DAO	depressive sx	BDI-II	3
[61] ([61])	PR	215	US	Undergraduate students in relationships ≥ 4 mo.	23.0	41	SD	NSSS-S, SDS	relationship sat	CSI	3
[103] ([103])	PR	1153	Portugal	Women in cohabiting monogamous relationships ≥ 1 yr.	31.6		SD	NSSS-S, SDI-2	relationship sat	GMREL	3
[350] ([350])	PR	619	US	Adults in relationships ≥ 6 mo. recruited from Mturk	38.0	79	S	NSSS	attachment	ECR	3
[36] ([36])	PR	97 (194)	Canada	Québécois mixed-sex couples undergoing fertility treatment	31.8		S	FSFI, IIEF	attachment	ECR-12	4
[258] ([258])	PR	268	US	Sexually active men in committed heterosexual relationships	37.3	71	SODE	SSS, IIEF	relationship sat	CSI	4
[28] ([28])	PR	145 (290)	Canada	Québécois mixed-sex couples	31.3	94	S	GMSEX	relationship sat	DAS-4	2
[207] ([207])	PR	317	Lebanon	Sexually active women with (*n* = 65) and without (*n* = 252) endometriosis	29.5		S	SSS-W	depressive sx & relationship sat	DASS, CSI	2
[193] ([193])	PR	237	Poland	Undergraduate students in heterosexual relationships ≥ 1 mo.	20.1		S	SSQ	relationship sat	CSI-32	2
[116] ([116])	PR	2754	Canada	Québécois university employees (43%) and students (57%)	37.2		S	1 item	depressive sx & relationship sat	PHQ-4, DAS-4	2
[60] ([60])	PR	1605	US	Adults from Mturk in romantic relationships ≥ 2 yr.	34.9	75	S	GMSEX	relationship sat & attachment	CSI-4, ECR	3
[301] ([301])	PR	79	Brazilian Amazon	Young adults (aged 18 to 35) from Quilombola communities (descended from African slaves)	23.6	1.3	SODPLE	FSFI, IIEF	physical health & life satisfaction	WHOQoL	34
[138] ([138])	DD	46,874	ISS: 43 countries	Women respondents in the International Sex Survey (ISS)	32.4		SD	GMSEX, SDI-2	only sex health *		1
[203] ([203])	PR	102 (204)	Turkey	Heterosexual married couples	30.1		S	NSSS	depressive sx	DASS-21	2
[273] ([273])	PR	1080	Spain	Women (aged 18 to 55) born in Spain or living in Spain ≥ 1 yr.	24.5		SO	NSSS, 1 item	only sex health *		1
[250] ([250])	PR	61 (122)	Australia	Women (aged 18 to 47) with endometriosis and their partners	30.7		S	NSSS	relationship sat	CSI-16	2
[344] ([344])	PR	4148	Norway	National population study of adults	46.5		S	1 item	relationship sat	1 item	1
[358] ([358])	PR	179 (358)	Iran	Heterosexual cohabiting married couples	37.0		S	1 item	relationship sat	1 CSI item	2
[239] ([239])	MT	282	Portugal	Sexually active (in previous month) heterosexual adults	28.9		SODPLE	FSFI, IIEF	depressive sx	EADS-21	9
[389] ([389])	PR	440	Poland	Sexually active, childless adults aged 18 to 40	24.3		S	SSQ	life satisfaction	SWLS	1
[291] ([291])	PR	274 (548)	Canada	Mixed-sex Québécois couples seeking relationship therapy	42.0		S	GMSEX	attachment	ECR	4
[221] ([221])	PR	543	Canada	French-Canadian community adults	36.2		S	GMSEX	attachment	ECR	2
[20] ([20])	PR	100 (200)	Saudi Arabia	Heterosexual Arabic married couples (half seeking treatment for sexual or marital problems)	33.4		S	ISS	relationship sat & attachment	DAS-4, ECR-R	6
[200] ([200])	PR	170	Iran	Married and sexually active patients with PTOD (n = 55) and healthy controls (n = 115)	38.4		S	SSSW	depressive sx & relationship sat	BDI, ENRICH	2
[87] ([87])	PR	211 (422)	Malawi	Heterosexual couples with at least one partner living with HIV	40.5	0	S	CSSS	relationship sat	1 CSI item	1
[290] ([290])	PR	102 (204)	US	Newlywed couples living in Northern Florida	30.9	76	S	ISS	relationship sat	QMI, SMD, KMSS	1
[154] ([154])	PR	88	US	People with epilepsy currently in relationships and on medication to control seizures	32.1	83	S	GMSEX	relationship sat, well-being	GMREL, AESMI	2
[97] ([97])	PR	442	Philippines	Heterosexual Christian adults in relationships	38.0		SD	GMSEX, 1 item	relationship sat	GMREL	3
[285] ([285])	PR	3890	US & Germany	Never married and divorced individuals who were currently single	39.9	77	S	1 item	life satisfaction	1 item	3
[77] ([77])	PR	125 (250)	Canada	Sexually active couples seeking treatment for Provoked VestibuloDynia (PVD)	28.1	73	S	GMSEX	relationship sat & attachment	CSI-32 & ECR-R	6
[330] ([330])	PR	1627 (3254)	US	Mixed-sex couples from the CREATE study	30.9	53	S	RELATE	relationship sat & attachment	CSI-4 & ECR-R	6
[119] ([119])	PR	113	Portugal	Women having completed treatment for cervical cancer	48.0		S	ISS	relationship sat & attachment	DAS-14 & ECR	3
[141] ([141])	PR	265	US & Canada	60–75 yo. adults in same-sex relationships	65.3	82	S	ISS	relationship sat	CSI-32	1
[309] ([309])	PR	171	Canada	Women in relationships who were 18–25 weeks pregnant with their first child	29.8	85	S	GMSEX	relationship sat	CSI-4	1
[63] ([63])	PR	820	Spain & Latin Am countries	Spanish-speaking adults in same-sex relationships ≥ 3 mo.	29.1		S	MGH-SFQ	relationship sat & attachment	GMREL & ECR-S	6
[257] ([257])	PR	303	US & Canada	Sexually active adults in cohabiting romantic relationships	35.9		S	GMSEX	relationship sat	PRQC	1

Note. PR = peer reviewed article; DD = doctoral dissertation; MT = master’s thesis; sxs = symptoms; sat = satisfaction; S = sexual satisfaction; O = orgasm; D = sexual desire; P = lack of pain; L = vaginal lubrication; E = erectile function. Sexual health scales: AESMI = Adult Epilepsy Self-Management Instrument; ASEX = Arizona Sexual Experience Scale; ASFI = AnorectalSexaul Function Index; BSFQ = Brief Sexual Functioning Questionnaire; CSFQ = Changes in Sexual Functioning Questionnaire; DISF = Derogatis Interview for Sexual Functioning; DSM-IV = Diagnostic and Statistical Manual of Mental Disorders—4th edition; FSDS-R = Female Sexual Distress Scale—Revised; FSFI = Female Sexual Function Index; FSQ = Female Sexual Quotient; GMREL = Global Measure of Relationship Satisfaction; GMSEX = Global Measure of Sexual Satisfaction; GRISS = Golombok-Rust Inventory of Sexual Satisfaction; GSF = Global Sexual Functioning; IIEF = International Index of Erectile Function; ISBI = Israeli Sexual Behaviour Inventory; ISS = Index of Sexual Satisfaction; MGH = Massachusetts General Hospital Sexual Functioning; MSFI = Male Sexual Function Index; MSHQ = Male Sexual Health Questionnaire; MSQ = Multidimensional Sexuality Questionnaire; MSSCQ-S = Multidimensional Sexual Self-Concept Questionnaire—Sexual Satisfaction; NSSS-S = New Sexual Satisfaction Scale—Short; ORS = Orgasm Rating Scale; PFSF = Profile of Female Sexual Function; PROMIS = Patient-Reported Outcomes Measurement Information System; PSSI = Pinnes Sexual Satisfaction Inventory; SFQ = Female Sexual Function Questionnaire; SII = Sexual Interaction Inventory; SIS = Sexual Interest and Satisfaction; SSFS = Short Sexual Functioning Scale; SSS-W = Sexual Satisfaction Scale for Women; SVQ = Sexual function-Vaginal changes Questionnaire—Sexual Functioning; WSIDI = Women’s Sexual Interest Diagnostic Interview. Correlate scales: BAI = Beck Anxiety Inventory; BDI = Beck Depression Inventory; BSI = Brief Symptom Inventory; CESD = Center for Epidemiologic Studies Depression Scale; CIDI = World Health Organization’s Composite International Diagnostic Interview; CSI = Couples Satisfaction Index; CSSS = Couples Sexual Satisfaction Scale; DAS = Dyadic Adjustment Scale; DASS = Depression Anxiety Stress Scales; DASS-21 = 21-item Depression Anxiety Stress Scales; ECR = Experiences in Close Relationships; EMS = ENRICH Marital Satisfaction Scale; EPDS = Edinburgh Postnatal Depression Scale; GDS = Geriatric Depression Scale; GHQ = General Health Questionnaire; GMREL = Global Measure of Relationship Satisfaction; HADS-D = Hospital Anxiety and Depression Scale; HAMD = Hamilton Rating Scale for Depression; KMSS = Kansas Marital Satisfaction Scale; LiSat = Life Satisfaction—life, relationship, & sexual satisfaction items; MAS = Marital Adjustment Scale; MASQ = Mood and Anxiety Symptom Questionnaire; MAT = Marital Adjustment Test; MGH-SFQ = Massachusetts General Hospital-Sexual Functioning Questionnaire; MHQ = Middlesex Hospital Questionnaire; MOQ = Marital Opinion Questionnaire; MSI-R = Marital Satisfaction Inventory—Revised; PANAS = Positive and Negative Affect Schedule; PHQ = Patient Health Questionnaire; PROMIS = Patient-Reported Outcomes Measurement Information System; PRQC = Perceived Relationship Quality Components—Global Relationship Quality; PRQC-S = Perceived Relationship Quality Components—Satisfaction; PSS = Perceived Stress Scale; QMI = Quality of Marriage Index; QOL = Heinrichs Quality of Life Scale; QSI = Quality of Sex Inventory; RAS = Relationship Adjustment Scale; SCL = 90-item Symptom Check List; SF-12 = 12-item Short-Form Health Survey; SF-36 = 36-item Short-Form Health Survey; SRADM = Shaheen-Rakhawy Anxiety & Depression Measure; STAI = State and Trait Anxiety Inventory; SWLS = Satisfaction with Life Scale; TSC-D = Trauma Symptom Checklist—Depression; WHO-5 = The World Health Organization-Five Well-Being Index; WHOQoL = The World Health Organization Quality of Life; ZAI = Zung Anxiety Index. An asterix (*) signifies that article contributed only correlations among sexual health indices. “---” = information not provided.

**Table 3 behavsci-15-01636-t003:** Associations between Sexual Health and both Individual and Relationship Functioning Correlates.

Category of Results								*SM Rel-Prob*	Funnel Plot Asym *Z*	Trim and Fill Adj. Est.	
Correlate Variable											
Sexual Health Variables	*k*	*N*	*r*	*SE*	95% *CI*	*Q*	*I* ^2^			*r*	*k*
Sexual health links to correlates											
Psychological distress											
Sexual satisfaction	69	17,234	**−.276 ******	.022	[−.318, −.233]	**4449.05 ******	91.27	.99	1.84		
Orgasms	97	58,706	**−.250 ******	.022	[−.294, −.207]	**13,035.00 ******	97.46	3.04	1.33		
Desire	64	13,722	**−.167 ******	.031	[−.228, −.105]	**4077.37 ******	94.54	3.20	−.12		
Lack of pain	42	8575	**−.183 ******	.047	[−.275, −.090]	**3230.86 ******	96.71	6.17	.22		
Lubrication	46	9090	**−.239 *****	.033	[−.304, −.174]	**1678.95 ******	92.78	4.19	1.35		
Erectile function	11	2127	**−.148 *****	.046	[−.238, −.059]	**24.95 ****	68.74	.92	−1.95		
Psychological well-being											
Sexual satisfaction	22	6719	**.343 ******	.042	[.260, .425]	**1041.04 ******	93.52	.22	−1.697		
Orgasms	23	9420	**.239 ******	.041	[.159, .320]	**601.83 ******	94.04	.32	−1.52		
Desire	15	3502	**.320 ******	.070	[.184, .457]	**354.57 ******	97.11	2.85	**−2.83 ****	**.335 ******	16
Lack of pain	12	3399	**.268 ******	.067	[.137, .399]	**215.11 ******	94.30	10.88	**−3.26 *****	**.336 ******	15
Lubrication	12	3399	**.196 ****	.063	[.072, .319]	**258.63**	90.94	2.23	−.72		
Erectile function	0	0									
Physical health											
Sexual satisfaction	12	1997	**.311 ******	.077	[.159, .462]	**464.55 ******	92.30	1.52	**−2.90 ****	**.387 *****	15
Orgasms	14	4160	**.278 ******	.066	[.149, .406]	**705.08 ******	95.00	.09	−1.52		
Desire	13	2073	**.270 ******	.068	[.137, .404]	**365.65 ******	89.62	1.50	**−2.35 ***	**.323 ******	15
Lack of pain	9	1588	**.249 ******	.073	[.106, .392]	**87.50 ******	82.16	11.09	**−2.44 ***	**.304 ******	11
Lubrication	9	1588	**.221 ***	.093	[.039, .402]	**146.89 ******	89.77	1.63	**−2.02 ***	**.221 ***	9
Erectile function	1	39									
Attachment anxiety											
Sexual satisfaction	22	10,156	**−.237 ******	.048	[−.331, −.142]	4575.31 ****	97.86	.96	**2.38 ***	**−.336 ******	30
Orgasms	9	3979	**−.149 ******	.022	[−.193, −.105]	13.26	40.04	.91	**−2.58 ****	**−.127 ******	12
Desire	5	1479	−.043	.046	[−.133, .048]	9.43	59.61	.86	−1.9		
Attachment anxiety											
Lack of pain	2	975	**−.145 ******	.031	[−.207, −.083]	.89	.00	.75			
Lubrication	2	975	**−.128 ******	.032	[−.190, −.067]	.22	.00	.75			
Erectile function	1	248									
Attachment avoidance											
Sexual satisfaction	22	10,156	**−.339 ******	.034	[−.405, −.272]	237.89 ****	93.38	.89	**3.51 *****	**−.339 ******	22
Orgasms	9	3979	**−.241 ******	.029	[−.270, −.158]	**24.02 ****	63.99	.91	−.19		
Desire	5	1479	**−.124 ****	.046	[−.213, −.034]	9.71 *	64.27	.86	**2.03 ***	**−.124 ***	5
Lack of pain	2	975	**−.165 ******	.040	[−.243, −.088]	1.58	36.82	.75			
Lubrication	2	975	**−.279 ******	.070	[−.416, −.412]	**5.56 ***	82.02	.75			
Erectile function	1	248									
Relationship satisfaction											
Sexual satisfaction	62	69,328	**.554 ******	.020	[.516, .592]	**4372.45 ******	97.83	.02	**−4.91 ******	**.589 ******	27
Orgasms	60	47,311	**.267 ******	.026	[.216, .317]	**7075.88 ******	97.55	.23	−1.37		
Desire	21	10,312	**.248 ******	.043	[.164, .331]	**1158.65 ******	95.78	1.16	−1.49		
Lack of pain	11	3976	**.183 *****	.053	[.079, .287]	**51.96 ******	88.48	1.18	.38		
Lubrication	14	6973	**.187 *****	.038	[.114, .261]	**90.39 ******	88.23	.99	.14		
Erectile function	1	248									
Links among sexual health dimensions											
Sexual satisfaction correlating with											
Orgasms	138	113,079	**.398 *****	.019	[.361, .435]	**24,581.54 ******	98.59	.17	**−1.99 ***	**.444 ******	156
Desire	56	70,791	**.382 *****	.029	[.325, .440]	**15,921.34 ******	98.84	.22	**−2.30 ***	**.467 ******	70
Lack of pain	26	11,834	**.375 ******	.053	[.271, .480]	**3540.22 ******	98.85	.31	**−2.67 ****	**.426 ******	29
Lubrication	29	12,318	**.462 ******	.043	[.379, .546]	**1995.12 ******	97.38	.05	**−3.41 *****	**.542 ******	36
Erectile function	13	6315	**.400 ******	.045	[.313, .488]	**193.87 ******	92.94	.43	1.36		
Orgasms correlating with											
Desire	56	21,958	**.339 ******	.027	[.287, .392]	**3555.92 ******	96.32	.25	**−3.19 ****	**.416 ******	71
Lack of pain	27	11,128	**.357 *****	.055	[.249, .465]	**3675.84 ******	98.65	.21	−1.26		
Lubrication	24	11,504	**.531 ******	.046	[.441, .621]	**2268.09 ******	99.12	.08	**−4.57 ******	**.531 ******	28
Erectile function	13	6541	**.527 ******	.032	[.465, .589]	**139.90 ******	90.87	.07	−.75		
Desire correlating with											
Lack of pain	28	11,240	**.265 ******	.039	[.189, .342]	**3568.58 ******	96.58	.45	−2.46 *	**.348 ******	36
Lubrication	28	11,804	**.393 ******	.034	[.326, .460]	**1929.99 ******	96.04	.03	−3.60 ***	**.464 ******	37
Erectile function	11	6183	**.343 ******	.058	[.230, .456]	**298.79 ******	96.06	1.04	−1.33		
Lack of pain correlating with											
Lubrication	25	10,422	**.555 ******	.037	[.482, .628]	**1447.75 ******	97.89	.04	**−3.17 ****	**.588 ******	28

Note. SM Rel-Prob = the relative probability of finding a contradictory finding (i.e., non-significant or in the opposite direction to a majority of the other effects for that pair of variables) compared to finding a consistent finding (estimated using the selection method analyses of [252]). Numbers close to 1 suggest no publication bias and numbers over 1 suggest a greater relative likelihood of non-significant or counter-intuitive findings getting published. Funnel Plot Asym *Z*= Eggers test of funnel plot asymmetry, another test of possible publication bias. Trim and fill methods were used to adjust the estimates of effects showing significant asymmetry. Significant effects have been bolded for ease of interpretation. N/A = These correlations are between constructs measured only in women or men but not in both genders (i.e., unique to either the FSFI or the IIEF but not common to both). **** *p* < .0001, *** *p* < .001, ** *p* < .01, * *p* < .05.

**Table 4 behavsci-15-01636-t004:** Results of Path Models on Meta-Analytic Correlation Matrices.

MODEL			99% CIs		MODEL			99% CIs	
Correlate Being Predicted					Correlate Being Predicted				
Predictors	*β*	*p*	LL	UL	Predictors	*β*	*p*	LL	UL
**PSYCH DISTRESS MODEL**					**WELL-BEING MODEL**				
SexH → psychological distress					SexH → well-being				
Sexual satisfaction	**−.175**	<.001	−.209	−.142	Sexual satisfaction	**.225**	<.001	.165	.285
Orgasms	**−.127**	<.001	−.161	−.093	Orgasms	**.087**	<.001	.026	.148
Desire	−.024	.054	−.055	.008	Desire	**.211**	<.001	.154	.268
Lack of pain	**−.030**	.025	−.064	.004	Lack of pain	**.169**	<.001	.108	.231
Lubrication	**−.065**	<.001	−.103	−.026	Lubrication	**−.131**	<.001	−.201	−.061
SexH → sex satisfaction					SexH → sex satisfaction				
Orgasms	**.163**	<.001	.134	.193	Orgasms	**.163**	<.001	.107	.219
Desire	**.204**	<.001	.177	.231	Desire	**.204**	<.001	.152	.256
Lack of pain	**.143**	<.001	.113	.173	Lack of pain	**.143**	<.001	.086	.199
Lubrication	**.216**	<.001	.182	.250	Lubrication	**.216**	<.001	.152	.280
Indirect paths					Indirect paths				
Orgasm → SexSat → Distress	**−.029**		−.036	−.021	Orgasm → SexSat → Well-being	**.037**		.021	.053
Desire → SexSat → Distress	**−.036**		−.044	−.027	Desire → SexSat → Well-being	**.046**		.029	.063
L-pain → SexSat → Distress	**−.025**		−.032	−.018	L-pain → SexSat → Well-being	**.032**		.017	.047
Lube → SexSat → Distress	**−.038**		−.047	−.029	Lube → SexSat → Well-being	**.049**		.029	.068
**PHYSICAL HEALTH MODEL**					**REL SATISFACTION MODEL**				
SexH → perceived health					SexH → rel satisfaction				
Sexual satisfaction	**.181**	<.001	.107	.255	Sexual satisfaction	**.559**	<.001	.518	.599
Orgasms	**.148**	<.001	.073	.223	Orgasms	**.104**	<.001	.063	.145
Desire	**.144**	<.001	.074	.214	Desire	**.058**	<.001	.019	.096
Lack of pain	**.129**	<.001	.053	.204	Lack of pain	.005	.734	−.036	.047
Lubrication	**−.069**	.039	−.155	.017	Lubrication	**−.152**	<.001	−.199	−.105
SexH → sex satisfaction					SexH → sex satisfaction				
Orgasms	**.163**	<.001	.095	.231	Orgasms	**.163**	<.001	.122	.205
Desire	**.204**	<.001	.142	.266	Desire	**.204**	<.001	.166	.242
Lack of pain	**.143**	<.001	.075	.211	Lack of pain	**.143**	<.001	.101	.185
Lubrication	**.216**	<.001	.139	.293	Lubrication	**.216**	<.001	.169	.263
Indirect paths					Indirect paths				
Orgasm → SexSat → Health	**.030**		.012	.047	Orgasm → SexSat → RelSat	**.091**		.067	.115
Desire → SexSat → Health	**.037**		.018	.056	Desire → SexSat → RelSat	**.114**		.091	.137
L-pain → SexSat → Health	**.026**		.010	.042	L-pain → SexSat → RelSat	**.080**		.056	.104
Lube → SexSat → Health	**.039**		.018	.060	Lube → SexSat → RelSat	**.121**		.093	.148
**ATTCH AVOIDANCE MODEL**					**ATTCH ANXIETY MODEL**				
Avoid & SexH → Sex sat					Attch anxiety → SexH				
Attch avoidance	**−.209**	<.001	−.280	−.139	Sexual satisfaction	**−.161**	<.001	−.230	−.092
Orgasms	**.136**	<.001	.055	.217	Orgasms	**.146**	<.001	.064	.228
Desire	**.205**	<.001	.130	.279	Desire	**.209**	<.001	.134	.284
Lack of pain	**.142**	<.001	.061	.223	Lack of pain	**.127**	<.001	.045	.209
Lubrication	**.172**	<.001	.079	.265	Lubrication	**.212**	<.001	.119	.304
Attch avoidance → SexH					SexH → sex satisfaction				
Orgasms	**−.241**	<.001	−.321	−.161	Orgasms	**−.149**	<.001	−.231	−.067
Desire	**−.124**	<.001	−.206	−.042	Desire	−.043	.179	−.125	.039
Lack of pain	**−.165**	<.001	−.246	−.084	Lack of pain	**−.145**	<.001	−.227	−.063
Lubrication	**−.279**	<.001	−.358	−.200	Lubrication	**−.128**	<.001	−.210	−.046
Indirect paths					Indirect paths				
Avoid → Orgasm → SexSat	**−.033**		−.055	−.010	Anxiety → Orgasm → SexSat	**−.022**		−.039	−.005
Avoid → Desire → SexSat	**−.025**		−.044	−.006	Anxiety → Desire → SexSat	−.009		−.027	.009
Avoid → L-pain → SexSat	**−.023**		−.041	−.006	Anxiety → L-pain → SexSat	**−.018**		−.034	−.003
Avoid → Lube → SexSat	**−.048**		−.077	−.019	Anxiety → Lube → SexSat	**−.027**		−.048	−.006

Note: Consistent with the EVSA and ASA conceptual models, sexual health was treated as a predictor for the correlates of psychological distress, well-being, physical health, and relationship satisfaction whereas attachment avoidance and anxiety were treated as predictors of sexual health. SexH = sexual health; SexSat = sexual satisfaction; L-pain = lack of pain; RelSat = relationship satisfaction; Attch = attachment; Avoid = attachment avoidance; Anxiety = attachment anxiety. All path models were run on correlation matrices of meta-analytic estimates using Mplus 7.11. The models were fully saturated and therefore gave perfect fit. Path coefficients significant at *p* < .01 have been bolded for ease of interpretation. CI = confidence interval; LL = lower limit; UL = upper limit.

**Table 5 behavsci-15-01636-t005:** Examining Moderation of Links between Orgasms and Correlates.

	Sexual Health Indicator									
Correlate Being Predicted	Sexual Satisfaction		Orgasms		Desire		Lack of Pain		Lubrication	
Moderators Tested	Estimate	*p*	Estimate	*p*	Estimate	*p*	Estimate	*p*	Estimate	*p*
Predicting sexual satisfaction										
intercept	---	---	**.391**	**<.0001**	**.382**	**<.0001**	**.438**	**<.0001**	**.536**	**<.0001**
Male contrast	---	---	**−.162**	**<.0001**	−.048	.479				
Age (centered at 38.9 years)	---	---	.001	.464	**.006**	**.032**	**.012**	**.019**	.006	.171
Clinical population contrast	---	---	**.130**	**.001**	.051	.406	−.028	.801	.003	.972
Peer-reviewed contrast	---	---	−.037	.527	.134	.214	−.184	.360	−.388	.087
Predicting psychological distress										
intercept	**−.311**	**<.0001**	**−.237**	**<.0001**	**−.133**	**.026**	**−.264**	**.004**	**−.280**	**<.0001**
Male contrast	.013	.816	.059	.290	.040	.641	---	---	---	---
Age (centered at 38.9 years)	−.001	.560	**−.006**	**.002**	−.002	.481	.001	.897	−.004	.116
Clinical population contrast	.068	.207	.014	.773	−.066	.404	.182	.127	**.169**	**.020**
Peer-reviewed contrast	−.009	.925	−.083	.322	.084	.584	−.165	.597	−.259	.228
Predicting psychological well-being										
intercept	**.349**	**<.0001**	**.191**	**.001**	**.374**	**<.0001**	**.336**	**.001**	.080	.408
Male contrast	−.199	.118	−.164	.223	−.177	.411	---	---	---	---
Age (centered at 38.9 years)	.006	.096	.003	.544	.008	.334	.005	.535	.009	.240
Clinical population contrast	−.040	.618	.043	.618	−.173	.219	−.153	.265	.181	.171
Predicting relationship satisfaction										
intercept	**.581**	**<.0001**	**.253**	**<.0001**	**.219**	**.004**	**.124**	**.046**	.122	.006
Male contrast	.047	.325	−.062	.331	−.042	.835	---	---	---	---
Age (centered at 38.9 yrs)	−.001	.640	.001	.668	.003	.450	**.008**	**.025**	.001	.846
Clinical population contrast	−.031	.463	.049	.384	.028	.778	.075	.421	**.143**	**.040**
Peer-reviewed contrast	−.152	.089	.015	.872	.066	.815	---	---	---	---

Note: Significant weighted meta-analytic regression coefficients have been bolded for ease of interpretation.

## Data Availability

All SPSS, R, and Mplus code and syntax is available on the osf.io listing for this project (https://osf.io/fgm72 (accessed on 25 April 2025)). SPSS, R, and Mplus datasets are also available upon reasonable request within that osf.io project.

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
