# Peer review of "Conceptual Frameworks Linking Sexual Health to Physical, Mental, and Interpersonal Well-Being: A Comprehensive Systematic Review and Meta-Analysis"

_behavsci, 2025, doi:10.3390/bs15121636_

Round 1

Reviewer 1 Report

Comments and Suggestions for Authors

Please see the attached word document with my notes regarding your manuscript.

Author Response

Dear Editors and Reviewers,

We would like to thank you for your thoughtful and supportive comments and suggestions for strengthening our manuscript. This comprehensive systematic meta-analytic review of the correlational findings in the broader sexual health literature will truly stand alone as a definitive work not only in its massive scope but also by offering conceptual frameworks to help guide future work. This is particularly true after we implemented your suggestions to update the review to include articles published in the last 5 years.

We would also like to thank you for your patience and understanding toward our revision process. It was no small undertaking to screen 700+ new titles and abstracts, read 250 new full-texts of records, extract data from 56 new records, and then rerun all analyses in the paper and update the tables and figures accordingly. Having done that, our meta-analytic review now incorporates data from close to a quarter million participants (N = 248,021), giving us greater confidence in the analyses presented.

The remainder of this response letter details the revisions we made for each comment and suggestion. Thank you again for providing such constructive and helpful suggestions. We truly appreciate not only your input but also the supportive and collegial tone with which it was given.

Sincerely,

The authors

EDITOR COMMENTS:

Dear authors,

I commend what you have done in the attached.

OUR RESPONSE: Thank you! It was a massive amount of work and we appreciate you seeing the potential in that work.

From a methodological point of view you searches finished in 2020 which for me is too far to identify this as current. Is there anyway you could search from 2020-2025?

OUR RESPONSE: We have now extended the search to include articles published up to September, 2025. We thank you for encouraging us to update our search!

Also considering the review is there anyway aspects like two reviewers undertaking the process could be achieved? Appreciate this request may not be possible because of resource.

OUR RESPONSE: The first and third authors screened the records and extracted the data as a collaborative team – checking each other’s work. Discrepancies were exceedingly rare (< 0.5%) and were resolved with discussion between the two coders. This gave us high levels of confidence in the resulting meta-analytic dataset.

To address this concern, we have added the following sentences to our “Data Collection Process” subsection of the methods:

“The data extraction process was conducted and checked independently by two of the authors. Discrepancies were rare (< 0.5%) and were resolved through discussion.”

I think you need to update the methods to reflect PRISMA 2020 statement see

URL: https://www.prisma-statement.org/prisma-2020-checklist

URL: https://www.prisma-statement.org/prisma-2020-flow-diagram

Best practice would be to include all the sub-headings in the order the come - you will note this means changes to the methods section. Please consider.

OUR RESPONSE: We have now reordered the methods section, using the subheadings from the PRISMA 2020 methods checklist in nearly the same order as they appear in that checklist. Thank you for helping us align our manuscript with the current best practices.

Reviewer 1 makes some excellent points that I agree with please carefully consider this review and take time to respond because I will value the reviewers comments at the resubmission stage. 

OUR RESPONSE: We agree with the editor that Reviewer 1 made some excellent suggestions. As detailed below, we carefully incorporated revisions aligned with those suggestions and thank Reviewer 1 for reading the manuscript so carefully.

Finally, given the novelty of your research it may be worth considering a pre-print, especially as there is not a protocol. I wish you the best for your changes and work.

OUR RESPONSE: Per your suggestion, we have posted a pre-print of this manuscript (as currently under review at Behavioral Sciences) to preprints.org. We thank you for this suggestion!

REVIEWER 1 COMMENTS:

 This meta-analysis explores a modified structure that implements both the Enduring Vulnerability Stress Adaptation model of relationships functioning and the Attachment System Activation model to incorporate and explore aspects of sexual health. The meta-analysis offered an array of compelling results, including the relationship between various facets of sexual health and overall functioning; moderation results similarly demonstrated interesting outcomes (!!): orgasms were associated with higher sexual satisfaction, particularly among women and people in clinical subsamples - and orgasms supported lower distress for older individuals as well. Thank you for allowing me to review this paper, I think that this work is especially compelling.

OUR RESPONSE: We thank Reviewer 1 for their kind words and for seeing the potential value in our work. Given the size and scope of the systematic review, massive amounts of time and effort have gone into crafting this manuscript. We are extremely grateful that Reviewer 1 appreciated our efforts.

A persistent concern I had in reading the introductory pieces was the lack of more recent literature. In some instances, there really may be a dearth of research in that specific area in the past five years, but I know this is not the case across the board (i.e., the orgasm gap). As a review, there is a need to have a comprehensive and up-to-date set of references; I am not sure I saw any that were more recent than 2019, which is now (somehow) six years ago. I think that the entire introduction needs to be updated accordingly and - while not absolutely essential - it may be within your best interest to consider expanding this review to contain updated research from the past ~4 years. I am worried this manuscript risks coming across as outdated.

OUR RESPONSE: We thank Reviewer 1 for encouraging us to expand our systematic review to include literature from the last 5 years. As mentioned above, we went back to the literature and comprehensively searched for relevant articles from the last 5 years, adding 77 new samples (for a total of 281) and roughly 300 new effects (for a total of over 1,200) to be meta-analyzed from a combined sample of close to a quarter million (N = 248,021) participants.

I would recommend finding more recent references for a couple of the segments of the introductory paragraph including the following points: (1) the sentence that starts ‘given the high value individuals..’

OUR RESPONSE: We agree and have added two citations from 2023 (one qualitative, one quantitative) to bolster that assertion:

Given the high value individuals place on orgasms (Opperman et al., 2014; Vail-Smith et al., 2023; Walker & Lutmer, 2023),

NEW CITATIONS

Vail-Smith, K., Chaney, B. H., & Williams, M. (2023). Differences in Important Components of Sexual Satisfaction as Identified and Experienced by Undergraduate Males and Females. American Journal of Sexuality Education, 18(3), 504-522. https://doi.org/10.1080/15546128.2022.2124211

Walker, A.M. & Lutmer, A. (2023). Caring, Chemistry, and Orgasms: Components of Great Sexual Experiences. Sexuality & Culture 27, 1735–1756. https://doi.org/10.1007/s12119-023-10087-x

(2) the second portion of that same sentence that ends in ‘mental health, and life satisfaction’.

OUR RESPONSE: We agree that the references for the second half of that sentence could benefit from being updated. We have now added two of the new citations contributing effects to the meta-analysis to the citations at the end of that sentence:

one line of research has focused specifically on links between orgasmic functioning and well-being, underscoring the importance of orgasms for sexual satisfaction, relationship satisfaction, physical health, mental health, and life satisfaction (e.g., Abramov, 1976; Brezsnyak & Whisman, 2004; Brody & Costa, 2009; Ellsworth & Bailey, 2013; Leavitt et al., 2021; Mangas et al., 2024).

NEW CITATIONS

*Leavitt, C. E., Leonhardt, N. D., Busby, D. M., & Clarke, R. W. (2021). When is enough enough? Orgasm’s curvilinear association with relational and sexual satisfaction. The Journal of Sexual Medicine, 18(1), 167-178. https://doi.org/10.1016/j.jsxm.2020.10.002

*Mangas, P., Sierra, J. C., & Granados, R. (2024). Effects of subjective orgasm experience in sexual satisfaction: A dyadic analysis in same-sex Hispanic couples. Journal of Sex & Marital Therapy, 50(3), 346-368. https://doi.org/10.1080/0092623X.2023.2295960  

There’s a bit of a discrepancy between the abstract iteration of the purpose of the current review and what is listed in the introduction (i.e., “the current review sought to…”). I was under the impression that sexual health was the core priority with orgasms being a subsidiary component, but in reading the intro it seems that orgasms and orgasmic functioning is the core focal point with sexual health themes being related to this topic as a series of outcomes. Please clarify to develop a more cohesive narrative between the abstract and intro (and anywhere else this may be mentioned). I think a reorganization of the intro paragraph is probably the easiest route - because overall it does seem to root into what the abstract synthesizes but isn’t especially clear.

OUR RESPONSE: We thank Reviewer 1 for bringing this slight discrepancy to our attention. We have now rewritten the first two sentences of the abstract to clarify the purpose of the current review and align that text with the opening paragraph of the introduction:

The current meta-analysis modified the Enduring Vulnerability Stress Adaptation model of relationship functioning and the Attachment System Activation model of individual functioning to incorporate various aspects of orgasmic functioning within the broader context of sexual health and sexual satisfaction. This provided conceptual frameworks for integrating the findings on a wide range of correlates of orgasms, sexual satisfaction, and other components of sexual health into comprehensive models of individual and interpersonal functioning to guide future research.”

- I think the mention of the clinical and non-clinical subsamples, first mentioned in section 1.1, should be mentioned as part of the introduction (I.e., opening paragraph). I think that a lot is covered in this work - and it is good work - but it should be listed comprehensively at the outset and then developed in each subsequent paragraph as appropriate.

OUR RESPONSE: We thank Reviewer 1 for bringing up another excellent suggestion. We have now added the following text to the end of the opening paragraph to introduce the idea of using meta-analytic regressions to test moderators of the links between sexual health and various aspects of well-being:

The current review extended previous research further by using a meta-analytical framework to synthesize and integrate previous findings, thereby allowing us to examine moderators of the links between various aspects sexual health and various forms of well-being. Given the gender disparities uncovered in the field of sexual health (e.g., Armstrong et al., 2012; Blair et al., 2018), gender was tested as a moderator. As sexual desire and performance vary with age (e.g., Gades et al., 2008; Twenge et al., 2017), age was also tested as a moderator of the salience of sexual health across the lifespan. Finally, given the reduced levels of sexual health observed among individuals with mental or physical disorders (e.g., Atarodi-Kashani et al., 2017; Chang et al., 2012), the type of population sampled within each study (i.e., clinical or nonclinical) was also examined as a moderator.

- Please include more updated references where available throughout the introduction (e.g., in the section labeled, ‘Orgasms as One Component of Sexual Health’).

OUR RESPONSE: We agree with this suggestion as well and have added updated references to the first two paragraphs of that section:

Orgasms as One Component of Sexual Health. A growing body of work has more specifically focused on orgasmic functioning as a key aspect of sexual health (e.g., Abramov, 1976; Brezsnyak & Whisman, 2004; Brody & Costa, 2009; Ellsworth & Bailey, 2013; Leavitt et al., 2021; Mangas et al., 2024). In fact, there is evidence suggesting that orgasms are still widely perceived as a key goal, if not the ultimate goal of sexual activity (e.g., Opperman et al., 2014; Vail-Smith et al., 2023; Walker & Lutmer, 2023), supporting their use as a marker of sexual health.”

- Orgasm gender gap section: there is more contemporary literature in this field; please include more recent research and references.

OUR RESPONSE: To address this we have added the following text to the end of the orgasm gender gap section:

Recent results in nationally representative samples continue to support an orgasm gender gap, potentially due to lower frequencies of clitoral stimulation in heteronormative sex (Andrejek et al., 2022; for a review see Andrejek et al., 2025).

NEW CITATIONS

Andrejek, N., Fetner, T., & Heath, M. (2022). Climax as work: Heteronormativity, Gender labor, and the gender gap in orgasms. Gender & Society, 36(2), 189-213. https://doi.org/10.1177/08912432211073062

Andrejek, N., Fetner, T., & Heath, M. (2025). Climax as work: Heteronormativity, Gender labor, and the gender gap in orgasms. In V. Taylor, L. J. Rupp, A. D. Crossley, & N. E. Whittier (Eds.) Feminist Frontiers: Readings on Gender, Sexuality, and Society, 194-211. Bloomsbury.

- Saying ‘see Figure 1A for full EVSA model’ implied this is the original framework of the model and not the proposed conceptual model - please clarify the language here. 2

OUR RESPONSE: We agree that clarification is needed. We now refer readers to Figure 1A with the following text:

see Figure 1A for the proposed conceptual model modifying the EVSA to include sexual health

- “Would buffer relationships from any possible adverse effects of difficulties with orgasms”: language used here is extreme, suggesting it eliminates any issues associated with orgasming - perhaps use less extreme language (e.g., drop the word ‘any’).

OUR RESPONSE: We thank Reviewer 1 for catching that extreme language. We have now dropped the word ‘any’ as suggested and have toned the language down a bit further:

The EVSA model would therefore suggest that adaptive processes like strong emotional support and healthy sexual communication could at least partially buffer relationships from the adverse effects of difficulties with orgasms

- The argument to explain the ASA model mapping onto sexual functioning is weak - at minimum there should be some references that support the assertions you’re making (e.g., the sentence that contains, ‘could be conceptualized as a clear threat, activating the attachment system…’).

OUR RESPONSE: We agree that assertion could be strengthened by a citation. That sentence now reads as follows:

From a sexual functioning perspective, orgasm difficulties and a failure to achieve an orgasm during a sexual encounter with a partner could be conceptualized as a possible threat (i.e., a failure of that masculinity achievement, see Chadwick & van Anders, 2017), activating the attachment system, and thereby promoting deactivation and hyperactivation strategies.”

NEW CITATION

Chadwick, S. B., & van Anders, S. M. (2017). Do women’s orgasms function as a masculinity achievement for men?. The Journal of Sex Research, 54(9), 1141-1152. https://doi.org/10.1080/00224499.2017.1283484

- “The current comprehensive literature search (described below) failed to identify any published systematic reviews or meta-analyses focused on examining the correlates of orgasms.” There are some more recent reviews that approach this topic, though perhaps not comprehensively, that should be considered (note that this does not detract from your work, but I think it’s important to address previous iterations that attempted to broach the topic, albeit incompletely or tangentially): - Social, behavioral, and psychological factors influencing the female orgasm: https://doi.org/10.47363/jsmr/2024(3)121 - Women’s orgasm and its relationship with sexual satisfaction and wellbeing (review): https://doi.org/10.1007/s11930-023-00371-0

OUR RESPONSE: We thank Reviewer 1 for pointing us toward those two recent reviews. We agree that the manuscript would be markedly strengthened by contextualizing the findings within the current review literature. In addition to the two reviews suggested by Reviewer 1, our comprehensive literature search uncovered another 4 recently published reviews relevant to the current review. As a result, we have modified the sentences opening that section of the introduction to more appropriately describe what is truly unique about the current review. We have also added language to that section describing those 6 new reviews:

Previous Reviews

               Narrative reviews. The current comprehensive literature search (described below) failed to identify any published meta-analytic systematic reviews with a comparably broad focus (i.e., exploring associations amongst components of sexual health and between those components and a range of factors representing individual and relationship functioning). However, the current literature search did uncover a number of published narrative reviews that focused on related topics. For example, White and Reamy (1982) published a narrative review of 74 articles examining sex during pregnancy, however none of their articles overlapped with the current sample of 173 records. Similarly, narrative reviews of female sexual functioning in old age (Wood et al., 2012; 3 of its 58 articles overlapped with the current review), the health benefits of various sexual activities (Brody, 2010; 9 of its 174 studies overlapped), the psychological and interpersonal correlates of sexual dysfunction (Brotto et al., 2016; 1 of its 364 articles overlapped), the links between sexual activity and both physical and mental health (Levin, 2007; 5 of its 74 articles overlapped), and links from women’s orgasms to well-being (Dienberg et al., 2023; 6 of its 85 studies overlapped) demonstrated similar low levels of overlap with the current review. Notably, although Meston and colleagues (2004b) published a narrative review of 323 articles examining women’s orgasms, the extremely broad scope of that review represented such a distinct focus that none of its articles overlapped with those in the current meta-analysis.

               Systematic reviews. The current comprehensive literature search also uncovered a number of relevant systematic reviews that have been published in the last 5 years. Given the distinct and slightly more narrow conceptual focuses of these reviews, they only demonstrated nominal overlap with the current review as they examined: (1) etiological factors shaping female sexuality (Ourania et al., 2024; 3 of its 21 studies overlapped), (2) predictors of sexual satisfaction (Rausch & Rettenberger, 2021; 6 of its 109 provided citations overlapped), (3) factors linked to sexual functioning in people living with HIV (Huntingdon et al., 2020; none of its 26 studies overlapped), and (4) factors linked to distress over lower sexual functioning (Stotz et al., 2025; 1 of its 19 studies overlapped). Finally, the current literature search uncovered a single meta-analytic systematic review with a related focus: examining various aspects of sexual communication and their links to sexual and relationship functioning (Mallory, 2022). However, given its primary focus on aspects of sexual communication, only 5 of its 93 studies overlapped with the current review.

Taken as a set, this overview of reviews of sexual health (and their markedly low levels of overlap with the current review) suggests that the current review offers a unique contribution to the current literature, integrating findings across a wide range of disparate fields and testing novel path models to evaluate unique (i.e., incremental) links from the various aspects of sexual health to the correlates examined. It also represents the first review to systematically examine the correlates of orgasmic functioning within the broader context of a multivariate conceptualization of sexual health, evaluating the unique links between orgasms and both individual and interpersonal functioning after controlling for other key aspects of sexual health.

NEW CITATIONS

Dienberg, M. F., Oschatz, T., Piemonte, J. L., & Klein, V. (2023). Women’s orgasm and its relationship with sexual satisfaction and well-being. Current Sexual Health Reports, 15(3), 223-230. https://doi.org/10.1007/s11930-023-00371-0

Ourania, P., Dimitra, M., Athina, D., & Victoria, V. (2024). Etiological factors affecting female sexuality: A systematic review. Journal of Sexual Medicine & Research, 3(1), 1-16.

https://doi.org/10.47363/JSMR/2024(3)121

Rausch, D., & Rettenberger, M. (2021). Predictors of sexual satisfaction in women: A systematic review. Sexual Medicine Reviews, 9(3), 365-380. https://doi.org/10.1016/j.sxmr.2021.01.001

Huntingdon, B., Muscat, D. M., de Wit, J., Duracinsky, M., & Juraskova, I. (2020). Factors associated with general sexual functioning and sexual satisfaction among people living with HIV: a systematic review. The Journal of Sex Research, 57(7), 824-835. https://doi.org/10.1080/00224499.2019.1689379

Stotz, T., Mackelprang, J. L., Harkin, A., Hunt, D., & Buzwell, S. (2025). When is low sexual function problematic for women? A systematic review of factors associated with distress about low sexual function. The Journal of Sexual Medicine, 22(10), 1827-1838. https://doi.org/10.1093/jsxmed/qdaf192

- Figure 3: this could be due to journal formatting but there are several components of this visual that are cut off (i.e., the n = portion of a few cells, as well as the end of the far right, bottom cell (n=3 separate papers with same [cut off]).

OUR RESPONSE: We will attempt to correct this when submitting the revision. We apologize for the inconvenience that caused and thank Reviewer 1 for their patience.

- “The majority of the (sub)samples having been published in the last 10 years (Table 1).” Related to what date? (e.g., published in the prior ten years related to the time of analysis).

OUR RESPONSE: As we have now updated the literature search (and all of our tables and figures) to include articles from the last 5 years, we have modified that sentence:

The (sub)samples had been published across a 54-year span, with a majority of the (sub)samples (77%) having been published in the last 15 years (Table 1).

- Table 3 and 4 are both cut off in the submission so cannot be thoroughly examined.

OUR RESPONSE: We will attempt to correct this when submitting the revision. We apologize for the inconvenience that caused and thank Reviewer 1 for their patience.

- The figures demonstrating unique links to variables of interest (e.g., psychological distress, physical health) located below Table 4: please make it clear which paths are significant in these models for ease of reading; I see the note at the bottom so maybe the Figure name can be adjusted to reflect this? Significant Results of Path Models etc. perhaps?

OUR RESPONSE: We have renamed Figure 4 as suggested, “Significant Results of Path Models…

- What is the argument for merging the clinical population together if there are so many unique conditions discussed in the record and manuscript characteristics (i.e., 33 distinct diagnoses)?

OUR RESPONSE: To ensure that this limitation and justification would be a the top of readers’ minds when reading these results, we now provide a justification for this choice at the beginning of the results section detailing those results:

Despite spanning over 33 distinct diagnoses, a majority of the conditions represented were more chronic in nature resulting in more chronic levels of impairment. Thus, we treated clinical vs. non-clinical populations as one of our moderators to be tested, collapsing across those individual disorders to focus on how impairment in health and individual functioning might impact the links examined.

- Biological mechanisms in the future directions section - more recent references please. Also there’s a review in that arena from 2012 I believe (https://doi.org/10.1016/j.neubiorev.2012.03.006). 3

OUR RESPONSE: We have added the suggested the recommended 2012 review as well as a more recent 2019 review to the opening of that section.

Biological Mechanisms. Although somewhat outside of the scope of the current review, a growing body of work has uncovered possible neurochemical mechanisms linking orgasms to emotional bonding (for reviews and greater details see Stoléru et al., 2012; Sayin & Schenck, 2019).”

NEW CITATIONS

Stoléru, S., Fonteille, V., Cornélis, C., Joyal, C., & Moulier, V. (2012). Functional neuroimaging studies of sexual arousal and orgasm in healthy men and women: a review and meta-analysis. Neuroscience & Biobehavioral Reviews, 36(6), 1481-1509. https://doi.org/10.1016/j.neubiorev.2012.03.006

Sayin, H. Ü., & Schenck, C. H. (2019). Neuroanatomy and neurochemistry of sexual desire, pleasure, love and orgasm. SexuS Journal Winter, 4(11), 907-946.

I read through the future directions twice – I don’t believe I have any comments there beyond noting that I appreciate the thorough nature of this portion of the manuscript.

OUR RESPONSE: We thank Reviewer 1 for these kind and supportive words.

REVIEWER 2 COMMENTS:

As the authors note, this meta-analytic review not only synthesizes quantitative findings but also offers concrete guidelines for expanding the past 49 years of research on the correlates of sexual health in a theoretically grounded manner

We find the article highly interesting, methodologically rigorous, and fully compliant with the necessary standards for publication.

OUR RESPONSE: We thank Reviewer 2 their kind words and for seeing value in our work.

Reviewer 2 Report

Comments and Suggestions for Authors As the authors note, this meta-analytic review not only synthesizes quantitative findings but also offers concrete guidelines for expanding the past 49 years of research on the correlates of sexual health in a theoretically grounded manner

We find the article highly interesting, methodologically rigorous, and fully compliant with the necessary standars for publication.

Author Response

REVIEWER 2 COMMENTS:

As the authors note, this meta-analytic review not only synthesizes quantitative findings but also offers concrete guidelines for expanding the past 49 years of research on the correlates of sexual health in a theoretically grounded manner

We find the article highly interesting, methodologically rigorous, and fully compliant with the necessary standards for publication.

OUR RESPONSE: We thank Reviewer 2 their kind words and for seeing value in our work.

Round 2

Reviewer 1 Report

Comments and Suggestions for Authors

I sincerely appreciate the time taken by the authors to integrate the suggestions made. I am comfortable with the updates and recommend it for publication.

Author Response

Dear Editor,

We are very grateful that you appreciate the effort and care we invested in our revisions. As requested, we have now moved the narration of the PRISMA diagram (and the diagram itself) to the beginning of the Results section. As suggested, we have also replaced the old diagram with one made with the free template.

We also noticed that part of Table 5 was missing and so we have now included the full Table 5. 

Thank you for a supportive and constructive review process. Please let us know if there is anything else you need.

Sincerely,

The authors